# Signaling mechanisms in renal compensatory hypertrophy revealed by multi-omics

Hiroaki Kikuchi [1] ✉, Chung-Lin Chou[1], Chin-Rang Yang[1], Lihe Chen[1], Hyun Jun Jung [2], Euijung Park[1], Kavee Limbutara[3], Benjamin Carter [4], Zhi-Hong Yang[5], Julia F. Kun[5], Alan T. Remaley[5] & Mark A. Knepper [1] ✉

Loss of a kidney results in compensatory growth of the remaining kidney, a phenomenon of considerable clinical importance. However, the mechanisms involved are largely unknown. Here, we use a multi-omic approach in a unilateral nephrectomy model in male mice to identify signaling processes associated with renal compensatory hypertrophy, demonstrating that the lipid-activated transcription factor peroxisome proliferator-activated receptor alpha (PPARα) is an important determinant of proximal tubule cell size and is a likely mediator of compensatory proximal tubule hypertrophy.

The kidney has a marked capacity for hypertrophy. When a single kidney is resected, a common event in the setting of kidney transplantation, renal trauma or renal cancer, the contralateral kidney undergoes an increase in size and function resulting in functional compensation (compensatory hypertrophy)[1,2]. The hypertrophy occurs at the level of individual renal tubules (nephrons), more than a million of which make up the renal parenchyma in humans[3,4]. Increases in renal tubule size are reported in the proximal tubule[5,6], the distal convoluted tubule[6] and collecting duct[6,7]. Nephron hypertrophy also occurs in another clinically important setting, viz., chronic kidney disease (CKD), where damaged nephrons undergo atrophy, but the remaining intact nephrons can undergo increases in size and function[8,9]. This response can blunt the initial decline in glomerular filtration rate (GFR), protecting the patient, but making CKD difficult to recognize in its early stages.

Compensatory increases in nephron size are preceded by hemodynamic changes, i.e., increased renal blood flow and single-nephron GFR[10,11]. With unilateral nephrectomy (UNx), GFR increases rapidly, i.e., within minutes or hours, well in advance of measurable increases in kidney mass. Nevertheless, it remains unclear whether the key signal converted to a cellular response is mechanical in nature, related to increased flow or pressure in the renal tubule, or could be a circulating factor, e.g a response to insulin-like growth factor or other growth factors[12,13]. Another key question is 'Do nephrons increase their size via increases in cell size (cellular hypertrophy), increases in cell number

(cellular hyperplasia) or both?', recognizing that the answer could be different in different renal tubular segments. As signaling mechanisms involved in cellular growth and cellular proliferation differ, an answer to this question is crucial to understanding the overall mechanism of renal compensatory hypertrophy.

Despite many reductionist investigations into the mechanisms of compensatory renal hypertrophy[13] suggesting several triggers such as mammalian target of rapamycin (mTOR) pathways[14–17] and growth factor signaling[18], our knowledge is incomplete, perhaps owing to the intrinsic complexity of the overall process for regulation of kidney size. However, in recent years, rapid progress has been made in the field of systems biology that is designed to decipher mechanisms of complex phenomena. The availability of ever-improving "-omic" methodologies is critical to this progress. Our laboratory has already employed multiple -omic approaches to the understanding of pathophysiological processes in the kidney, for example, the syndrome of inappropriate antidiuresis (SIADH)[19] and lithium-induced nephrogenic diabetes insipidus (NDI)[20].

Here, we use an array of -omic approaches (quantitative proteomics, RNA-seq based transcriptomics, Assay of Transposase Accessible Chromatin sequencing (ATAC-seq) and phospho-proteomics) to identify key processes in the renal proximal tubule associated with compensatory hypertrophy, showing that the lipid-activated transcription factor, peroxisome proliferator-activated receptor alpha (PPARα),

[1]Epithelial Systems Biology Laboratory, Systems Biology Center, National Heart, Lung, and Blood Institute, National Institutes of Health, Bethesda, MD, USA. [2]Division of Nephrology, Department of Medicine, Johns Hopkins University School of Medicine, Baltimore, MD, USA. [3]The Center of Excellence in Systems Biology, Faculty of Medicine, Chulalongkorn University, Bangkok, Thailand. [4]Laboratory of Epigenome Biology, Systems Biology Center, National Heart, Lung and Blood Institute, NIH, Bethesda, MD, USA. [5]Lipoprotein Metabolism Section, Translational Vascular Medicine Branch, National Heart, Lung and Blood Institute, National Institutes of Health, Bethesda, MD, USA. ✉e-mail: hiroaki.k1114@gmail.com; knepperm@nhlbi.nih.gov

is an important determinant of proximal tubule cell size and is a likely mediator of compensatory proximal tubule hypertrophy.

## Results

### Animal model and hypotheses

The objective was to make multi-omic observations at 24-h and 72-h time points in the contralateral kidney after the left kidney was resected or sham surgery was performed in male mice (Fig. 1a), and then mine the data to identify signaling pathways involved in the hypertrophic response. Measurements of kidney weight over body weight ratio (KW:BW) showed rapid growth of the contralateral kidney, not matched after sham surgery (Fig. 1b and Supplementary Fig. 1a) (See also Supplementary Fig. 1b and Supplementary Data 1). The maximum KW:BW ratio was seen by day 3, indicating that relevant gene expression changes that trigger the hypertrophy likely occur within the first 3 days and that substantial growth is already seen within the first 24 h. Figure 1c shows representative kidney sections 3 days after unilateral nephrectomy or sham surgery. Histology of kidney sections indicated increased thickness of the renal cortex and increased coronal length of the kidney (Fig. 1d and Supplementary Fig. 1c for definitions of length and Supplementary Data 2 for individual data). Figure 1e shows confocal fluorescence images of proximal tubules (top) and cortical collecting ducts (bottom) that were microdissected 30 days after sham surgery or UNx, revealing morphological effects on these tubule segments. Morphometry of the microdissected tubules (Fig. 1f) revealed significant increases in outer diameter and mean cell volume in proximal tubules, but no clear increase in the cell count per unit length (See also Supplementary Fig. 1d, Supplementary Fig. 1e). In contrast, in cortical collecting ducts, there was a significant increase in both outer diameter and cell count per unit length (Fig. 1g, See Supplementary Data 3 for individual data). Overall, these observations indicate that (a) compensatory hypertrophy after unilateral nephrectomy occurs rapidly, i.e. in the time frame of 0–3 days, (b) that the response occurs not only in the proximal tubule but also in the collecting duct, and (c) that the increase in proximal tubule diameter occurs largely as a result of increased cell volume.

The increase in cell size in proximal tubules presumably involves both transcriptional and posttranscriptional regulation of anabolic processes. A list of hypotheses about signals resulting from nephrectomy that could be transduced to trigger these changes is provided in Fig. 2, Supplementary Data 4 and the Supplementary Discussion 1. These hypotheses summarize signaling pathways known to mediate responses to mechanical or metabolic signals likely to be triggered by loss of one kidney. In the following, we carry out multi-omic analysis of the response of the contralateral kidney to unilateral nephrectomy at 24 and 72 h following surgery, consisting of ATAC-seq and RNA-seq of microdissected proximal tubules, as well as proteomic and phosphoproteomic analysis to discover the mechanisms involved.

### DNA accessibility and transcriptomics

Two methods can be employed to identify candidate transcription factors, namely (a) ATAC-seq to measure enrichment of transcription factor binding motifs at promoters and enhancers[21]; and (b) RNA-seq to identify transcription factors known to be expressed in the first portion (S1 Segment) of the proximal tubule.

We have recently reported comprehensive transcriptomic profiling of all 14 nephron segments microdissected from mice[22]. Supplementary Table 1 shows the most abundant transcription factors in the S1 segment with reference to our hypotheses about signal transduction in compensatory hypertrophy (from Fig. 2 and Supplementary Data 4). Note that many of these candidate transcription factors are in the category "Lipid-Sensing Nuclear Receptors".

Assessment of chromatin accessibility by ATAC-seq analysis in microdissected S1 proximal tubules at 24 h after surgery is shown in Fig. 3 (see Supplementary Fig. 2a, b and c for quality control

information). These data are made available to readers as individual tracks on a genome browser at https://esbl.nhlbi.nih.gov/IGV_mo/ (place gene symbol into second box). Figure 3a shows changes in chromatin accessibility in UNx relative to sham treatment. Accessibility ratios are plotted against mean peak heights across all samples. The peaks that were significantly upregulated or downregulated (Differentially accessible regions; DAR) are indicated in pink (Benjamini-Hochberg FDR < 0.05). Of the 125,973 total detected peaks, 4223 (3.4%) were significantly altered in the UNx proximal tubules vs. sham. To identify the changes in chromatin accessibility that were most likely to be relevant for gene regulation, we cross-referenced our peak set with annotated enhancer and promoter regions. To investigate whether specific transcription factor pathways were associated with increased chromatin accessibility in the UNx samples, we performed motif enrichment analysis using the set of peaks with significantly increased DNA accessibility in the UNx proximal tubules (Fig. 3b). Highly enriched are binding-site motifs corresponding to Hepatocyte nuclear factor-4 alpha (HNF4α) and Peroxisome proliferator-activated receptor alpha (PPARα), two lipid-regulated transcription factors. Other lipid-regulated nuclear receptors from Supplementary Table 1 (Farnesoid X receptor (FXR), liver X receptor alpha (LXRα) and liver X receptor beta (LXRβ)) did not exhibit motif enrichment in UNx relative to sham. Also, motifs associated with the glucocorticoid receptor NR3C1 was similarly unenriched in UNx vs. Sham (See Supplementary Data 5 for full results of motif enrichment). Importantly, the HNF4α/PPARα binding motif exhibited sharply elevated enrichment in the differentially accessible peak set compared to the total peak set (47% vs. 28%), while no enrichment is found for the Hepatocyte nuclear factor-1 beta (HNF1β) motif (Fig. 3c, Supplementary Data 5), indicating that the HNF4α/PPARα motif is abundant in proximal tubules at baseline and this motif is further enriched in UNx vs. Sham. Motif analysis of peaks that exhibited decreased accessibility revealed only one significant match to known transcription factor motifs, namely HNF1β (Supplementary Fig. 2d, Supplementary Data 5 for full motif analysis data). Full ATAC-seq data can be viewed in Supplementary Data 6 or at https://esbl.nhlbi.nih.gov/IGV_mo/. Next, we used the ATAC-seq data for all detected peaks to perform transcription factor footprinting for PPARα to visualize the relationship between PPARα motifs and chromatin accessibility. We observed a well-defined footprint immediately surrounding PPARα motifs (Fig. 3d-Top). UNx samples showed increased chromatin accessibility surrounding PPARα motifs, while UNx samples did not show any difference of chromatin accessibility surrounding CTCF motifs (Fig. 3d-Bottom). These data suggest that occupancy of the PPARα binding motif is elevated with UNx treatment, suggesting a role for the PPARα pathway in the response to UNx.

To investigate the effect of UNx specifically at gene regulatory elements, we examined the change of chromatin accessibility at promoter-transcription start sites (TSS). Figure 3e shows a volcano plot for ATAC-seq data including only promoter-associated peaks (located within −1000 to +100 bp relative to TSS). Note that there are more promoters that show increases in ATAC-seq signals than show decreases. Volcano plots for promoter-TSS associated peaks for PPARα (Fig. 3f) and HNF4α (Supplementary Fig. 2e) show significant upregulation of DNA accessibility at PPARα and HNF4α target genes in the UNx treatment group.

To address possible signal transduction pathways highlighted in Supplementary Table 1, we carried out statistical analysis of mean ATAC-seq peak heights (promoter-TSS region only) for genes that are associated with each transduction pathway. Curated target genes for each transcription factor are summarized in Supplementary Data 7. At 24 h, the largest changes of the average of log$_2$ (UNx/Sham) for all identified regions can be seen for NR1H4, PPARα and HNF4α target genes (Supplementary Fig. 2f) (See Supplementary Data 8 for statistics). Figure 3g shows three examples of ATAC-seq peaks for known genes regulated by PPARα and HNF4α (Supplementary Fig. 2g), each

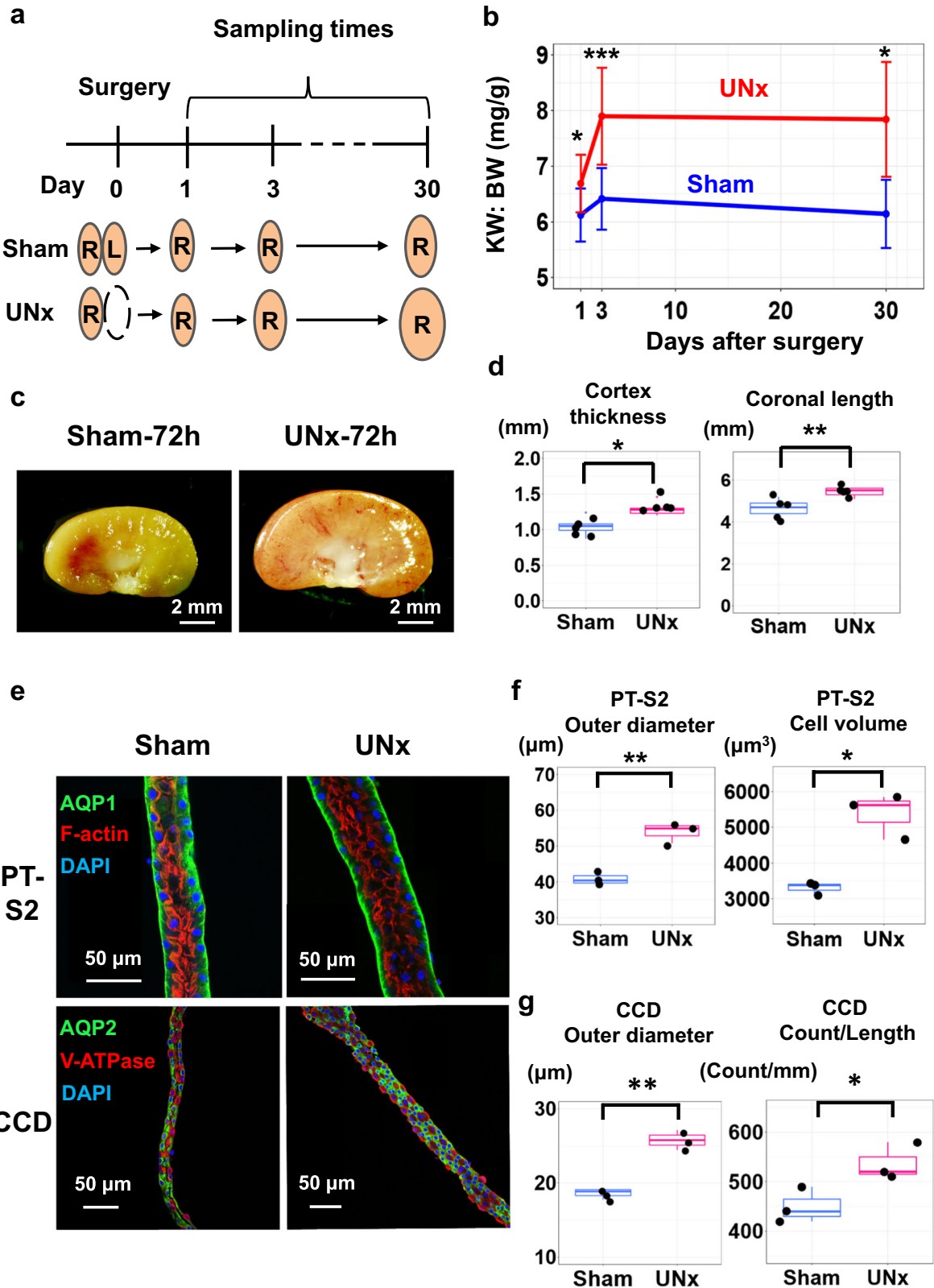

showing differences in peak heights in their promoter-TSS regions (DAR: highlighted in green).

## RNA-seq in microdissected proximal tubules at 24 h

Next, we used RNA-seq in newly prepared S1 proximal tubules microdissected from contralateral mouse kidneys at 24 h time point after UNx or sham surgery to identify known transcription factor target genes that undergo altered expression. All data are available on a Shiny-based web page (https://esbl.nhlbi.nih.gov/UNx/).

Figure 4a summarizes the RNA-seq data for S1 proximal tubules 24 h after surgery (See also Supplementary Fig. 3a, Supplementary Data 9). 215 transcripts were significantly increased ($p_{adj} < 0.05$ and $\log_2[UNx/Sham] > 0.50$) and 95 transcripts were significantly decreased ($p_{adj} < 0.05$ and $\log_2[UNx/Sham] < -0.50$) in

**Fig. 1 | Compensatory hypertrophy after unilateral nephrectomy (UNx) occurs rapidly and is largely because of increased cell volume in proximal tubule.**
**a** Sample collection protocol. R, right kidney; L, left kidney. **b** Time course of kidney growth. ($n = 11$ for sham-day 1, $n = 12$ for sham-day 3, $n = 5$ for sham-day 30, $n = 9$ for UNx-day 1, $n = 15$ for UNx-day 3, $n = 5$ for UNx-day 30). Data are presented as mean ± SD. KW, kidney weight; BW, body weight. *$P < 0.05$; ***$P < 0.001$ (two-sided Student's $t$ test). **c** Representative kidney images for Sham and UNx mice 3 days after surgery. **d** Kidney size parameters. Data are presented as mean ± SD. *$p < 0.05$, **$p < 0.01$ (Cortex: $p = 0.011$, Coronal: $p = 0.005$, two-sided Student's $t$ test). **e** Representative confocal fluorescence images of microdissected proximal tubules (S2 segment, PT-S2) and cortical collecting ducts (CCD) from Sham and UNx mice.

PT-S2, straight part of proximal tubule obtained from medullary ray in cortex region. Representative images were selected from $n = 3$ sham vs. UNx sets. **f** Size metrics for PT-S2 as calculated by IMARIS image analysis software. PT measurements of tubule outer diameter and cell volumes were significantly elevated in UNx (pink) vs. Sham (blue) samples. Morphometry method described in Supplementary Fig. 1e. Data are presented as mean ± SD. **$p = 0.003$, *$p = 0.031$, two-sided Student's $t$ test. Box-and-whisker plots represent median and 25th and 75th percentiles-interquartile range; IQR and whiskers extend to maximum and minimum values. **g** Size metrics for CCD as calculated by IMARIS image analysis software. Morphometry described in Supplementary Fig. 1e. **$p = 0.0015$, *$p = 0.033$, two-sided Student's $t$ test. Box-and-whisker plots as in (**f**).

unilaterally-nephrectomized mice versus sham out of a total of 13,296 identified transcripts. Thus, transcript abundance changes were seen in only a small fraction of transcripts, specifically 2.3% of total, suggesting a precise physiological response that may be indicative of the initial trigger for the resulting hypertrophy. Of interest, many of the upregulated transcripts are involved in PPARα-dependent lipid sensing and metabolism (highlighted in red in Fig. 4a), consistent with the ATAC-seq results. For example, HMG-CoA synthase (*Hmgcs2*), coding for the mitochondrial fate-committing ketogenic enzyme, was highly upregulated in UNx ($p_{adj} = 0.02$ and $\log_2[\text{UNx/Sham}] = 3.35$). Free fatty acids induce *Hmgcs2* expression in a PPARα-dependent manner[23]. Another is *Cyp4a10*, which mediates conversion of arachidonate to 20-hydroxyeicosatetraenoic acid (20-HETE), an important lipid mediator in the kidney[24]. Additionally, angiopoietin-like 4 (*Angptl4*), negatively regulated by AMP-activated protein kinase (AMPK), is markedly increased[25].

We performed Gene-Set Enrichment Analysis (GSEA) to examine whether differentially expressed genes were enriched for particular biological roles (Fig. 4b, Supplementary Table 2). The most highly enriched biological process term was seen for "FATTY ACID METABOLISM". Also highly enriched were "MTORC1 SIGNALING", "OXIDATIVE PHOSPHORYLATION" and "CHOLESTEROL HOMEOSTASIS", which are all relevant to the hypothesized mechanisms (Supplementary Data 4). De-enriched sets included "MITOTIC SPINDLE" and "G2M_CHECK POINT", consistent with the observed lack of proliferative response (Supplementary Fig. 1d). Chi-square analysis of the GSEA data revealed that, in contrast to the "FATTY ACID METABOLISM" and "MTORC1 SIGNALING", the oxidative phosphorylation biological pathway was not enriched upon UNx (Supplementary Fig. 3b). To address possible signal transduction pathways highlighted in Supplementary Table 1, we carried out statistical analysis of transcript abundances for target gene sets that are associated with each pathway (Fig. 4c). Large increases can be seen for PPARα and HNF4α target genes, and small decreases were seen for STAT6, SMAD4, CREM, and NR1H4 target genes (Supplementary Data 8). Ingenuity (IPA) upstream regulator analysis of regulated transcripts (Fig. 4d, Supplementary Fig. 3c) showed that the top 3 transcription factors predicted to be activated are PPARα, SREBF1, and HNF4α, and kinases predicted to be activated are serine/threonine kinase 11 (STK11, also called LKB1), and insulin receptor (INSR) (Full analysis in Supplementary Data 10). Plotting the relationship between ATAC-seq data (promoter-TSS region) and RNA-seq data for PPARα target genes showed a highly significant correlation (Fig. 4e), but a significant correlation was not seen for target genes of HNF4α (Fig. 4f). Furthermore, no significant relationship was found between ATAC seq peaks in distal enhancer regions (Intergenic and Intronic regions) and either PPARα and HNFα target genes (Supplementary Fig. 3d). These data are consistent with a transcriptionally activating role for PPARα at gene promoters during UNx-induced tubular hypertrophy.

## RNA-seq in cortical collecting duct
Consistent with prior studies[5,6], we confirmed that cortical collecting ducts (CCD) undergo an increase in size in response to nephron loss

(Fig. 1e, g). Therefore, RNA-seq in microdissected cortical collecting ducts was performed in order to compare with proximal tubules. (https:/esbl.nhlbi.nih.gov/UNx/). Figure 5a shows a volcano plot summarizing the RNA-seq data at 24 h. In contrast to the proximal tubule, the data show increases in a large number of genes associated with cell proliferation including E2F target transcripts ($p_{adj} < 0.05$ and $\log_2$ UNx/Sham > 0.50, highlighted in red). (see also Supplementary Fig. 4a, Supplementary Data 11). Figure 5b shows GSEA analysis indicating a significant increase in genes associated with "E2F_TARGETS", "G2M_CHECKPOINT", "MYC_TARGETS", and "MITOTIC_SPINDLE" all consistent with a proliferative response. Significantly downregulated was "TNFA_SIGNALING_VIA_NFKB"(See also Supplementary Table 3). Cell cycle-associated transcripts that were increased in response in the UNx collecting ducts are shown in Supplementary Fig. 4b. The general picture at 72 h was the same as that seen at 24 h (Fig. 5c, Supplementary Fig. 4c, d, Supplementary Table 3; see Supplementary Data 12 for full data). Consistent with a proliferative response, Ki-67 labelling was markedly increased in collecting duct principal cells at 72 h (Fig. 5d, Supplementary Fig. 4e).

## RNA-seq in microdissected proximal tubules at 72 h
Next, we used RNA-seq (Fig. 6a, Supplementary Fig. 5a) in S1 proximal tubules microdissected from contralateral mouse kidneys at a later point (72 h), at which time the UNx-induced hypertrophy had plateaued. Figure 6a shows a volcano plot summarizing the data (See Supplementary Data 13 for full data). There were only 73 transcripts that were increased and 11 transcripts that were decreased. In contrast to the result at the 24-h timepoint, GSEA identified "E2F_TARGETS" and "G2M_CHECKPOINT" as the genes sets with the top two normalized enrichment scores (NES), both pointing to the cell cycle and its regulation (Fig. 6b, Supplementary Table 2). Volcano plot for GSEA E2F target genes consistently shows significant upregulation of cell cycle related genes (Fig. 6c). Upregulated genes associated with "G2M checkpoint" and "E2F targets" in GSEA substantially overlapped with those seen in CCD at the 24 h timepoint, pointing to a tendency toward a hyperplasia profile despite the lack of morphological evidence for frank proximal tubule hyperplasia (Fig. 6d, Supplementary Data 14, Supplementary Fig. 1d). Adjusted $p$ values and $\log_2$ ratios for curated E2F target genes (Fig. 6e-left), PPARα target genes (Fig. 6e-right), and curated GSEA target genes for "FATTY ACID METABOLISM", "MTORC1 SIGNALING", and "OXIDATIVE PHOSPHORYLATION" (Supplementary Fig. 5b) are visualized in bubble plots showing differential changes between the 24 and 72 h timepoints. Upregulation of PPARα target genes, "FATTY ACID METABOLISM" related genes, "MTORC1 SIGNALING", and "OXIDATIVE PHOSPHORYLATION" related genes at 24 h become less evident at 72 h, while that of E2F target genes become more evident. However, similar to the 24-h time point, IPA upstream analysis for transcription factors suggested activation of SREBF1, PPARα and HNF4α (Fig. 6f, Supplementary Data 15). Thus, there appears to be an effect on the cell at 72 h superimposed on the growth response seen at 24 h. The result of IPA upstream analysis for kinases at 72 h was consistent with that at 24 h showing insulin receptor (INSR) and STK11 (LKB1) as the top activated kinases (Supplementary Fig. 5c).

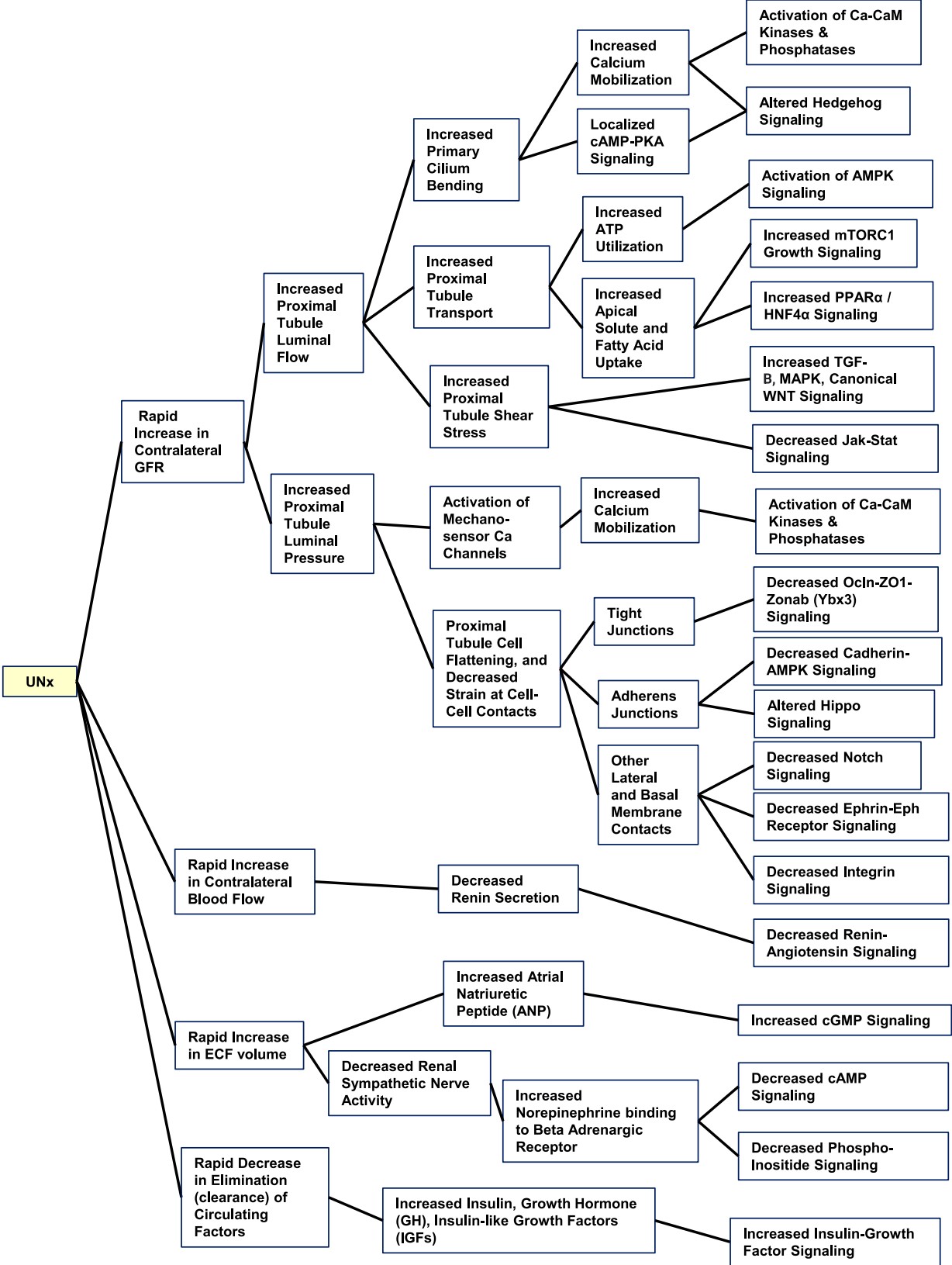

**Fig. 2 | Hypotheses regarding possible signaling mechanisms in contralateral kidney triggered by unilateral nephrectomy.** These hypotheses summarize signaling pathways known to mediate responses to mechanical or metabolic signals likely to be triggered by loss of one kidney. (See also Supplementary Discussion 1 and Supplementary Data 4). UNx unilateral nephrectomy, GFR glomerular filtration rate, ECF extracellular fluid, Ca calcium, CaM calmodulin, TGF transforming growth factor, OCLN occludin.

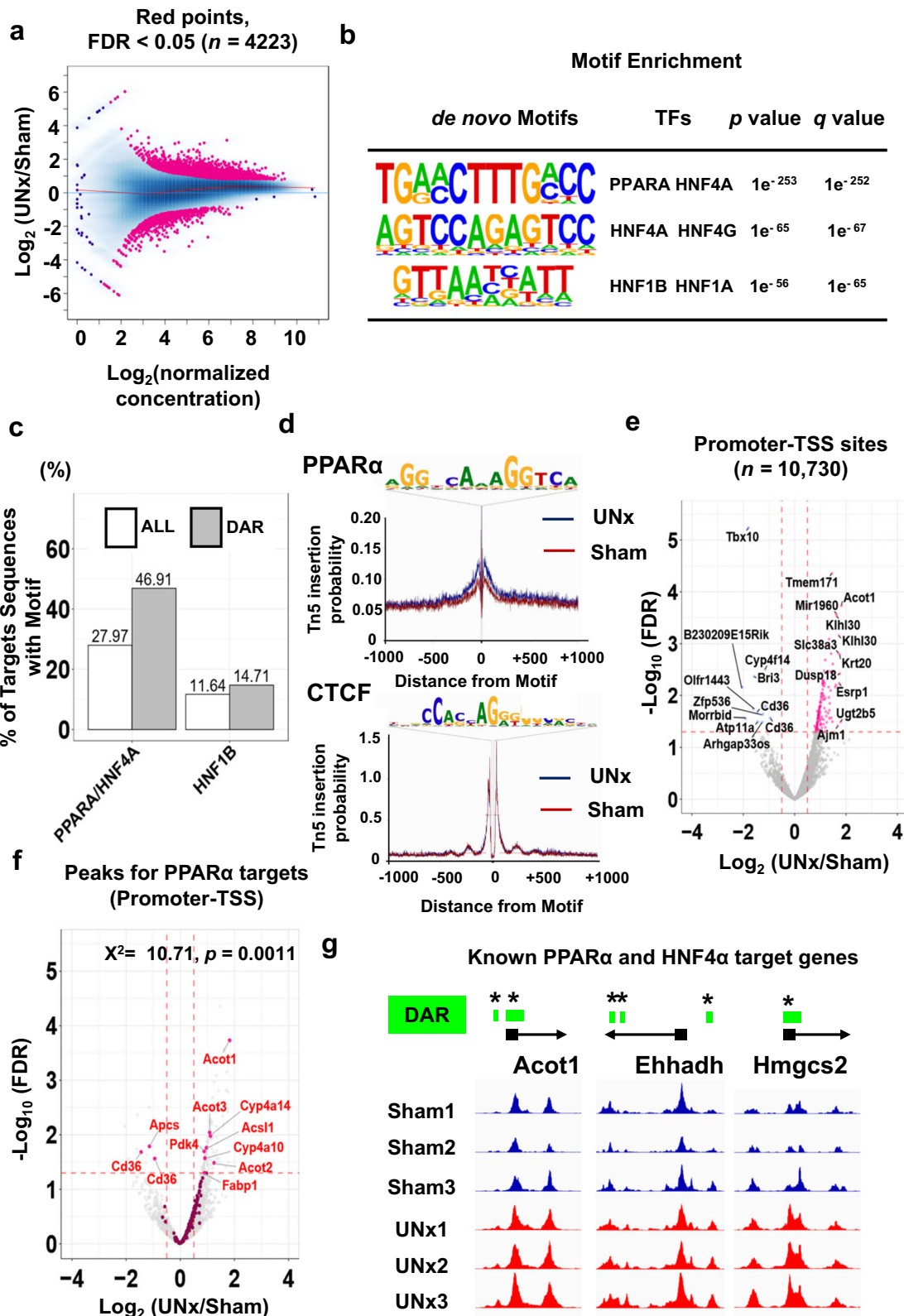

## Proteomic response to unilateral nephrectomy at 24 h

Transcriptomics is considerably more sensitive than proteomics, but measurement of protein responses is necessary to validate conclusions from transcriptomics and to understand mechanisms. Therefore, we carried out proteomics profiling using TMT-based quantitative protein mass spectrometry at 24 h and 72 h after UNx or sham surgery. To maximize profiling depth, these studies were done in whole-kidney samples, recognizing that about 65% of kidney protein in mouse is from the proximal tubule[26]. These data can be viewed at https://esbl.nhlbi.nih.gov/Databases/UnX-proteome/index.html and https://esbl.nhlbi.nih.gov/Databases/UnX-Phospho/72hlog.html.

The 24-h time point was studied to explore regulatory responses. Of the 4881 proteins quantified, 128 were increased and 39 were decreased (Fig. 7a) based on dual criteria (dashed lines, $p < 0.1$ and $\log_2$

**Fig. 3 | Single tubule ATAC-seq for microdissected proximal tubules from Sham and UNx at the 24 h timepoint. a** Ratio-intensity plot. Points represent peak regions determined by MACS2 for individual peaks. The x-axis represents average signal intensity within the region, and the y-axis represents $\log_2$ ratio of Sham and UNx signals. Red line is Loess fit. Among the 125,973 open chromatin regions identified among all samples, 4223 were identified as differentially accessible sites (FDR < 0.05, highlighted in magenta). ($n = 3$ for each group) **b** HOMER analysis identifies the enriched TF binding motifs in chromatin regions that are more accessible in UNx vs Sham (applying cumulative hypergeometric distribution adjusted for multiple testing with the Benjamini-Hochberg method) (See also Supplementary Data 5 for all results). **c** Percentages of motif sequences in all peaks (white) or in differentially accessible regions (grey) for the PPARα/HNF4α target motif and the HNF1B target motif. **d** Transcription factor foot-printing profiles generated using all identified ATAC-seq peaks for PPARα and CTCF (negative control) for microdissected PT-S1 to quantitate Tn5 insertion enrichment in the UNx vs. sham. **e** Volcano plot indicating peaks of chromatin accessibility within annotated promoter-TSS regions. Magenta points, increased accessibility in the UNx vs Sham treatment (FDR < 0.05 and $\log_2$(UNx/Sham)> 0.5). Blue points, decreased accessibility (FDR < 0.05 and $\log_2$(UNx/Sham) < −0.5). Chromatin accessible regions with top 10 or bottom 10 $\log_2$(UNx/Sham) values annotated by nearest gene name. **f** Volcano plot indicating chromatin accessibility at promoter regions near annotated TSS. Magenta points, PPARα target regions. Target genes for PPARα are listed in Supplementary Data 7. **g** Examples of ATAC-seq peaks for known PPARα and HNF4α target genes, each showing differences in peak heights between UNx and Sham treatments at their promoter-TSS regions (highlighted in green; DAR, differentially accessible region, vertical axes are of equal length). All data are available on a genome browser at https://esbl.nhlbi.nih.gov/IGV_mo/ and a Shiny-based web page (https://esbl.nhlbi.nih.gov/UNx/). * FDR < 0.05 (Benjamini and Hochberg method, adjusted $p$ values are provided in Supplementary Data 6).

UNx/Sham > 0.20 or < −0.20) (See also Supplementary Data 16). PPARα target proteins with significant changes are labeled in red. (The $\log_2$ UNx/Sham threshold of [−0.2,0.2] defines a more than 99 % confidence interval based on sham vs. sham comparisons.) Volcano plots for PPARα target proteins and for GSEA targets identified as activated in RNA seq at 24 h show significant upregulation of target proteins for PPARα targets (Fig. 7b-left), "MTORC1 SIGNALING", (Fig. 7b-center), "FATTY ACID METABOLISM" (Supplementary Fig. 6a) but not for "OXIDATIVE PHOSPHORYLATION" (Fig. 7b-right). Although a correlation between proteomics data (24 h) and RNA-seq data (24 h) for all identified proteins was not evident (Fig. 7c), a significant correlation was seen for PPARα target gene products (Fig. 7d), giving a protein-level validation of the PPARα transcriptomic response. On the other hand, there were no statistically significant correlations for MTOR target proteins (GSEA) nor OXPHOS target proteins (GSEA) (Supplementary Fig. 6b).

IPA upstream analysis for transcription factors at 24 h showed that PPARα was among the top 3 predicted transcription factors in the regulation of protein expression, while HNF4α was not (Fig. 7e, Supplementary Data 17 for full analysis). Based on IPA upstream analysis for kinases at 24 h, AMPK (PRKAA 1, 2) was predicted to be inhibited, while AKT1 was predicted to be activated at 24 h (Supplementary Data 17). Testing these hypotheses, immunoblots showed a decrease in phosphorylation of AMPK at an activating site (T172) (Supplementary Fig. 6c), and an increase in phosphorylation of AKT at an activating site (S473) (Supplementary Fig. 6d).

To confirm the results above in proximal tubule-enriched samples, proteomics was also done in samples in which kidney cortex was dissected out in lieu of whole kidney analysis. (Supplementary Fig. 6e). Proximal tubules account for 84% of cortex vs. 65% of whole kidney[26]. As shown in Supplementary Fig. 6f, substantial de-enrichment of collecting ducts (AQP2) were found in kidney cortex compared to whole kidney samples. Of the 5778 proteins quantified, 208 were increased and 352 were decreased (Fig. 7f) based on dual criteria (dashed lines, $p < 0.1$ and $\log_2$ UNx/Sham > 0.20 or <−0.20) (See also Supplementary Data 18). IPA upstream analysis for transcription factors using kidney cortex samples showed PPARα as the top predicted transcription factor in the regulation of protein expression, which was consistent with the result from the whole kidney (Fig. 7g, Supplementary Data 19). IPA upstream analysis for kinases showed, in contradiction to that from whole kidney analysis, AMPK (PRKAA 1, 2) as activated, and MTOR as inhibited (Supplementary Fig. 6g, Supplementary Data 19).

## Proteomic response to unilateral nephrectomy at 72 h
72-hour proteomics (Fig. 8a) showed that of the 5855 proteins quantified, 165 were increased and 594 were decreased based on the same criteria as used for the 24-hour data (Full data in Supplementary Data 20).Volcano plots (Fig. 8b) and bubble plots (Fig. 8c) for PPARα target proteins, GSEA targets for "FATTY ACID METABOLISM" and "MTORC1 SIGNALING" show that upregulation of these target proteins found at 24 h become less evident at 72 h, consistent with the results from trajectory analysis using RNA seq datasets. In contrast to the result from RNA seq at 72 h, there was no significant increase in "E2F_TARGETS" proteins (Fig. 8d). IPA upstream analysis for transcription factors predicted that both PPARα and HNF4α were activated (Fig. 8e-left), and IPA analysis for kinases suggested that AMPK (PRKAA 1, 2) may be activated (Fig. 8e-right) at 72 h (Supplementary Data 21).

## Phosphoproteomic response to unilateral nephrectomy
Quantitative phosphoproteomics analysis was carried out for whole kidneys after UNx or sham surgery at two time points, namely 24 and 72 h. (See https://esbl.nhlbi.nih.gov/Databases/UnX-Phospho/ and Supplementary Data 22 and 23 for full data). Both 24 h and 72 h phosphoproteomics data point to downregulation of AMP-regulated kinase (AMPK) activity. Specifically, phosphorylation of Prkab1 (AMPK beta-1 regulatory subunit) at S108 is known to induce AMPK activity[27] and phosphorylation at this site was decreased at the 72 h time point ($\text{Log}_2$ (UNx/Sham) = -0.67, Supplementary Data 24). This result fits with the conclusion that AMPK is downregulated from immunoblotting of AMPK phosphorylated at T172 showing a decrease (Supplementary Fig. 6c). Another hypothesis was that increased growth factor signaling could be involved in the hypertrophic response. This hypothesis appears to be ruled out by two results at 72 h showing:: (a) a strong decrease in phosphorylation of ERK1 (Mapk3) $\text{Log}_2$(UNx/Sham) = −0.58 and (b) a strong decrease in phosphorylation at S244 of Pdpk1 (3-phosphoinositide-dependent protein kinase 1) ($\text{Log}_2$(UNx/Sham) = −0.95). Finally, we hypothesized that mammalian target of rapamycin (mTOR) signaling could be involved in compensatory hypertrophy (Fig. 2, Supplementary Table 4 and Supplementary Discussion). Phosphoproteomic analysis at 72 h showed a strong decrease in phosphorylation of mTOR at S2448, consistent with a decrease in mTOR enzyme activity[28].

## Multi-omics data integration
We integrated results from nine -omics data sets presented in this study (Fig. 9a) using the Database for Annotation, Visualization and Integrated Discovery (DAVID) (https://david.ncifcrf.gov/home.jsp) to identify biological processes most likely to regulate renal hypertrophy in proximal tubules (see Supplementary Data 25). Gene ontology analysis predominantly highlighted changes in fatty acid metabolism including "lipid transport", which is consistent with PPARα activation, as concluded above (Fig. 9b). Among genes/proteins included as "Lipid Transport", Cd36, Fabp5, Atp8a1, Atp11c, Gm2a, Gramd1b, Npc2, Apoa2, Apoa4, Atg9a, Cert1, Osbpl9, Slc10a2 and Tex2 were identified as changed in our data integration analysis. In contrast to lipid transporters, $\log_2$ values for amino acid transporters reported to be expressed in the kidney[29] were curated from RNA seq and proteomics data at both time points. Among them, only

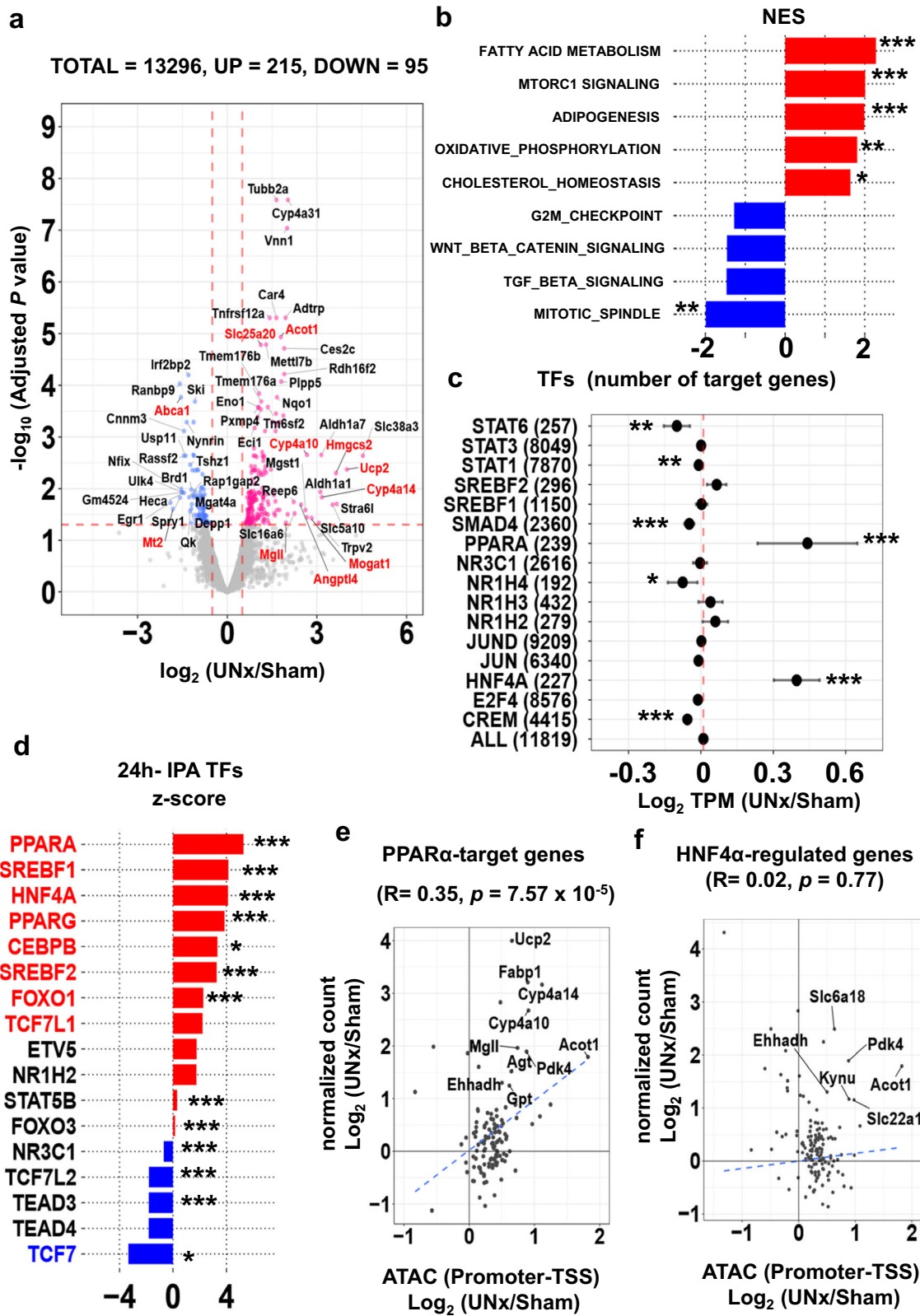

the glutamine transporter Slc38a3 showed significant increase at 24 h (Supplementary Data 26).

**Lipid analysis**

PPARα endogenous ligands are saturated and unsaturated fatty acids or their derivatives including eicosanoids and arachidonic acid metabolites[30,31]. Among these, polyunsaturated fatty acids (PUFA) are

especially known to be potent ligands for PPARα[32]. To determine the endogenous ligands of PPARα which are enriched in the kidneys from UNx compared to sham, we measured kidney fatty acid levels using gas chromatography (GC) (Supplementary Data 27). As shown in Fig. 9c, the concentrations of total saturated fatty acids (SAT FA), mono-unsaturated fatty acids (MUFA), n-6 and n-3 PUFA, and total PUFA were increased significantly by 25.6%, 67%, 34.8%, 23.5% and 29.5%,

**Fig. 4 | Single-tubule RNA-seq for microdissected S1 proximal tubules (PT-S1) from Sham and UNx mice transcripts at 24 h. a** Volcano plot. Magenta points, significantly increased expression. Blue points, significantly decreased expression. Significant differential expression was determined using thresholds of $p_{adj} < 0.05$ and $|\log_2(UNx/Sham)| > 0.5$. Red font, genes known to be regulated by PPARα. ($n = 3$ for Sham, $n = 4$ for UNx). **b** Top-ranked Hallmark Pathway gene sets determined using Gene Set Enrichment Analysis (GSEA). Normalized enrichment score (NES) was calculated using PT-S1 differentially expressed genes for UNx vs. Sham treatments at 24 h post-surgery. *$p < 0.05$, **$p < 0.01$, ***$p < 0.001$ (weighted Kolmogorov–Smirnov test, p values are provided in Supplementary Table 2). **c** Target gene set analysis for TFs listed in Supplementary Table 1 at 24 h after UNx. $\log_2$(UNx/Sham) values were plotted for members of curated target gene sets. Error bars indicate 95% confidence interval. The measure of center for the error bars is the averages of $\log_2$ ratios (UNx/Sham) of TPM values (in RNA-seq) for TF-target gene sets. The gene sets were listed in Supplementary Data 6. *$p < 0.05$, **$p < 0.01$, ***$p < 0.001$ (un-paired, two-tailed Student's $t$-test, specific p values provided in Supplementary Data 8.) ($n = 3$ for Sham, $n = 4$ for UNx). **d** Prediction of upstream regulatory transcription factors using Ingenuity Pathway Analysis (IPA). Predictions performed using differentially expressed genes in UNx vs. Sham treatments at 24 h. Genes with normalized z-scores larger than 2, red; those with normalized z-scores less than −2, blue. *$p < 0.05$, **$p < 0.01$, ***$p < 0.001$. Fisher's exact test (specific p values provided in Supplementary Data 10). **e**, **f** Correlation between gene expression and chromatin accessibility (UNx vs. sham treatments) for PPARα target genes (**e**) and for HNF4α target genes (**f**) at 24 h timepoint. *$p < 0.05$, **$p < 0.01$, ***$p < 0.001$ (Pearson's correlation). TSS, transcription start site. Significant correlation was assessed with Pearson's product moment correlation coefficient using the stat_cor function (method = Pearson, two-sided) in R.

respectively, in the kidney from UNx compared with the Sham group. This constitutes a significant increase of the levels of endogenous ligands of PPARα in the kidney after UNx treatment, suggesting a potential mechanism of PPARα activation in this context. In addition, the quantitative analysis of lipid content in mouse kidney (Fig. 9d and e, Supplementary Data 27) revealed that the concentration of triglycerides and phospholipid in the kidney from UNx group were increased by ~2-fold and 80.8%, respectively, compared with Sham group, which is consistent with the increased levels of overall fatty acid concentrations in UNx group.

### PPARα regulates cell size in the renal proximal tubule

The above multi-omics analyses suggests that PPARα may be involved in responses to renal hypertrophy following UNx surgery. However, it is not clear whether the activation of PPARα is the result or cause of renal hypertrophy, or both. Accordingly, we tested whether either activation of PPARα or deletion of PPARα affects cell size. To test whether PPARα activation alters proximal tubule cell size, we administered the PPARα agonist fenofibrate (50 mg/kg BW i.p. daily) or a control vehicle for 14 days (Supplementary Fig. 7a). Fenofibrate administration strongly increased the transcript abundance of PPARα target genes *Cyp4a10, Cyp4a14, Acox1* and *Acot1* in liver (Supplementary Fig. 7b), and *Acot1* and *Hmgcs2* in kidney (Fig. 10a), overlapping regulated genes seen by RNA-seq and ATAC-seq in UNx (Fig. 3g, Fig. 4a). Fenofibrate-treated mice had significantly increased kidney weight (KW) and KW:BW (Fig. 10b, Supplementary Fig. 7c). Cortical thickness was also significantly increased (Figs. 10c and 10d, Supplementary Data 28). Confocal fluorescence images of proximal tubules (Fig. 10e) microdissected from vehicle-treated mice (Left) versus fenofibrate-treated mice (Right) showed significant increases in outer diameter and mean cell volume in response to fenofibrate (See also Supplementary Data 29). A cellular size increase was also seen in liver, suggesting hypertrophy caused by fenofibrates is systemic, rather than specific for the kidney (Supplementary Fig. 7d, Supplementary Data 30).

To determine whether genetic deletion of PPARα affects proximal tubule cell size, we carried out studies in PPARα-null mice[33] (Fig. 10f, Supplementary Fig. 7e). We observed that selected genes, that are significantly upregulated in the proximal tubules in UNx at 24 h were significantly downregulated in kidneys of PPARα-null mice (Fig. 10g, Supplementary Fig. 7f). The kidneys appear to be smaller prior to unilateral nephrectomy (Fig. 10h, Supplementary Fig. 7g) and were substantially smaller 3 days after unilateral nephrectomy (Fig. 10i, Supplementary Fig. 7h). Thus, we conclude that PPARα is an important determinant of kidney size. Confocal images of microdissected proximal tubules showed a clear decrease in the cellular volume and tubular outer diameter, and an increase in the cell count per unit length in PPARα-null mice (Fig. 10j, Supplementary Fig. 7i, Supplementary Data 31), showing that PPARα is an important determinant of proximal tubule cell size.

## Discussion

Application of three different -omic methods (ATAC-seq, RNA-seq, and quantitative protein mass spectrometry) in addition to kidney tissue lipid analysis, identified PPARα in the proximal tubule as being associated with compensatory growth of proximal tubule cells in response to unilateral nephrectomy. PPARs represent a family of ligand-activated nuclear hormone receptors belonging to the steroid receptor superfamily[34]. Although the -omics studies provide strong evidence that PPARα is activated in the proximal tubule after unilateral nephrectomy, they did not establish whether PPARα could have a causal role in proximal tubule cell growth. To address this possibility, we performed two types of experiments. First, we showed the PPARα activation through the administration of fenofibrate resulted in an increase in kidney size associated with a marked increase in proximal tubule cell size. Second, deletion of the PPARα gene in mice resulted in a markedly diminished kidney size. Therefore, we conclude that PPARα plays a critical role in determination of kidney size, a finding that fits with the multi-omics data pointing to a key role for PPARα in compensatory kidney growth. Here, we discuss some issues that arise from our findings in light of existing literature.

Kidney weight normalized by body weight was already increased at 24 h after UNx, reaching a plateau at 72 h. We report that, at early stages of compensatory growth of the kidney, proximal tubule mass increases chiefly via increased proximal tubule cell size, rather than cell number, a finding consistent with previous studies[35,36]. Given that single-nephron GFR increases within minutes or hours with UNx[11,12], and that lipid ligands for PPARα in kidney tissue accumulated by 24 h after UNx, it is likely that the initial activation of PPARα occurs earlier than 24 h post-UNx. In contrast to the proximal tubule, the cortical collecting duct appears to exhibit compensatory growth through an increase in cell number. RNA-seq of CCDs revealed a strong pattern of increase in the abundance of cell-cycle associated transcripts after UNx, consistent with a proliferative response, confirmed through Ki-67 labelling. Thus, the mechanism of compensatory growth is not uniform along the renal tubule.

Cell enlargement requires the production of new cell membranes. Cell membranes are composed of glycerophospholipids, sphingolipids and sterols[37], as well as cardiolipin in mitochondria[38]. These components can be increased in the cell in several ways including transport into the cell, de novo synthesis in the cell, reduced shedding[39], repurposing of available membrane[40] and reduced lipid degradation. Based on the RNA-seq data, mechanisms for lipid transport and de novo synthesis appear to be activated in the proximal tubules after UNx treatment (https://esbl.nhlbi.nih.gov/UNx/). In particular, mRNA levels of two important lipid transporters, *Cd36* and *Slc27a2*, were substantially increased. Beyond this, mRNA encoding the lipid export transporters *Abca2* and *Abca5*, were significantly decreased. In terms of lipid synthesis, mRNA encoding several synthetic enzymes were increased, especially those regulated by SREBP1, a TF that controls expression of a wide array of genes involved in biosynthesis of

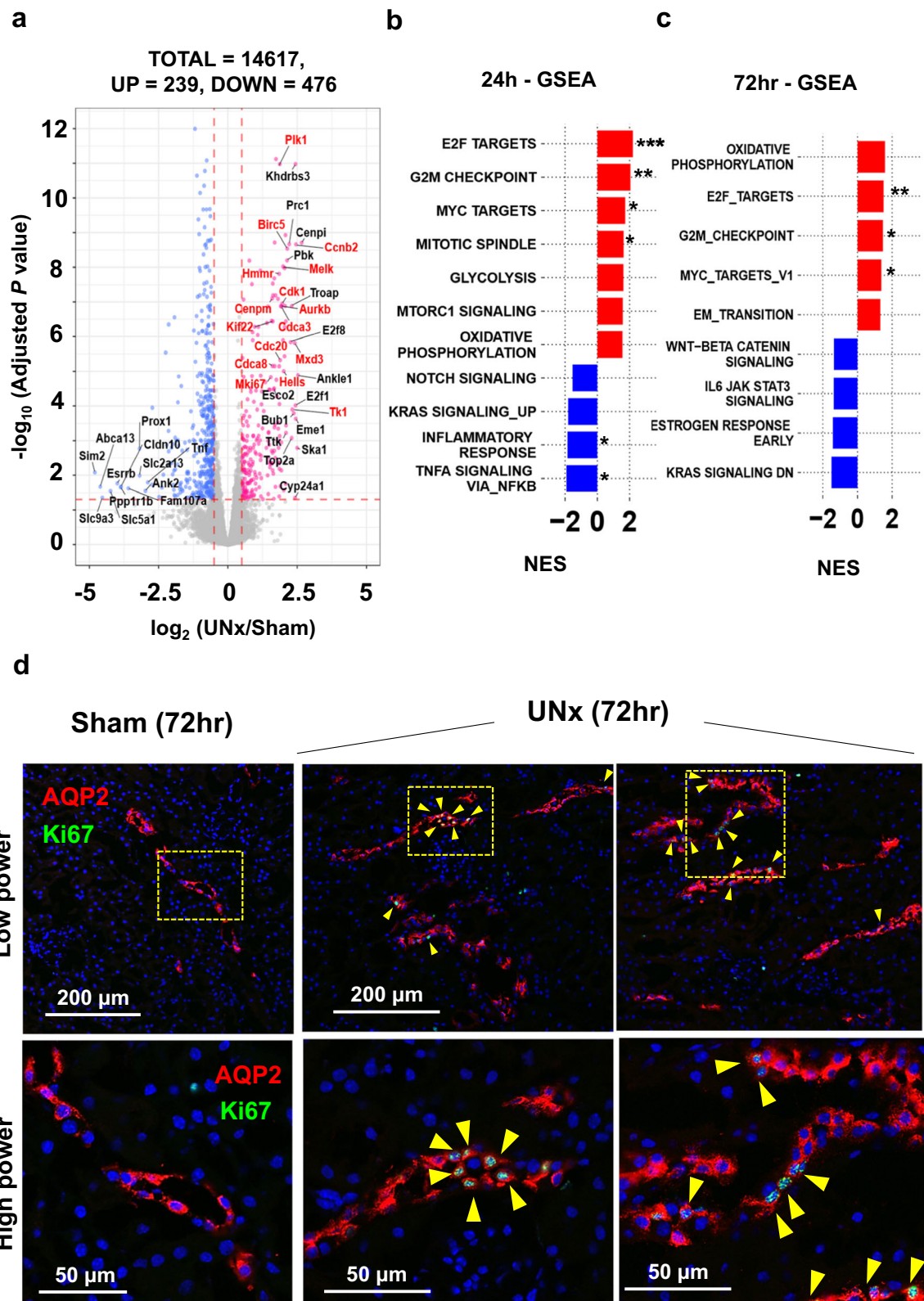

**Fig. 5 | Single-tubule RNA-seq for microdissected cortical collecting duct (CCD) from Sham and UNx at the 24 h and 72 h timepoint. a** Volcano plot for UNx vs. Sham. Magenta points, significantly increased expression. Blue points, significantly decreased expression. Significant differential expression was determined using thresholds of $p_{adj} < 0.05$ and $|log_2(UNx/Sham)| > 0.5$. Known E2F targets highlighted in red font. ($n = 4$ for Sham, $n = 3$ for UNx). **b, c** Top-ranked Hallmark Pathway gene sets determined using Gene Set Enrichment Analysis (GSEA). Normalized enrichment score (NES) calculated using CCD differentially expressed genes for UNx vs. Sham treatments at 24-h post-surgery (**b**) and at 72-h post-surgery (**c**). *$p < 0.05$, **$p < 0.01$, ***$p < 0.001$ (weighted Kolmogorov–Smirnov test, $p$ values are provided in Supplementary Table 3). d Immunofluorescence labeling of mouse renal cortex 72 h after Sham or UNx. Labelling for aquaporin-2 (AQP2; red) identifies collecting ducts. Ki67 (green) identifies dividing cells (yellow arrows). Representative images were selected from Sham ($n = 8$) and from UNx ($n = 7$) biological replicates.

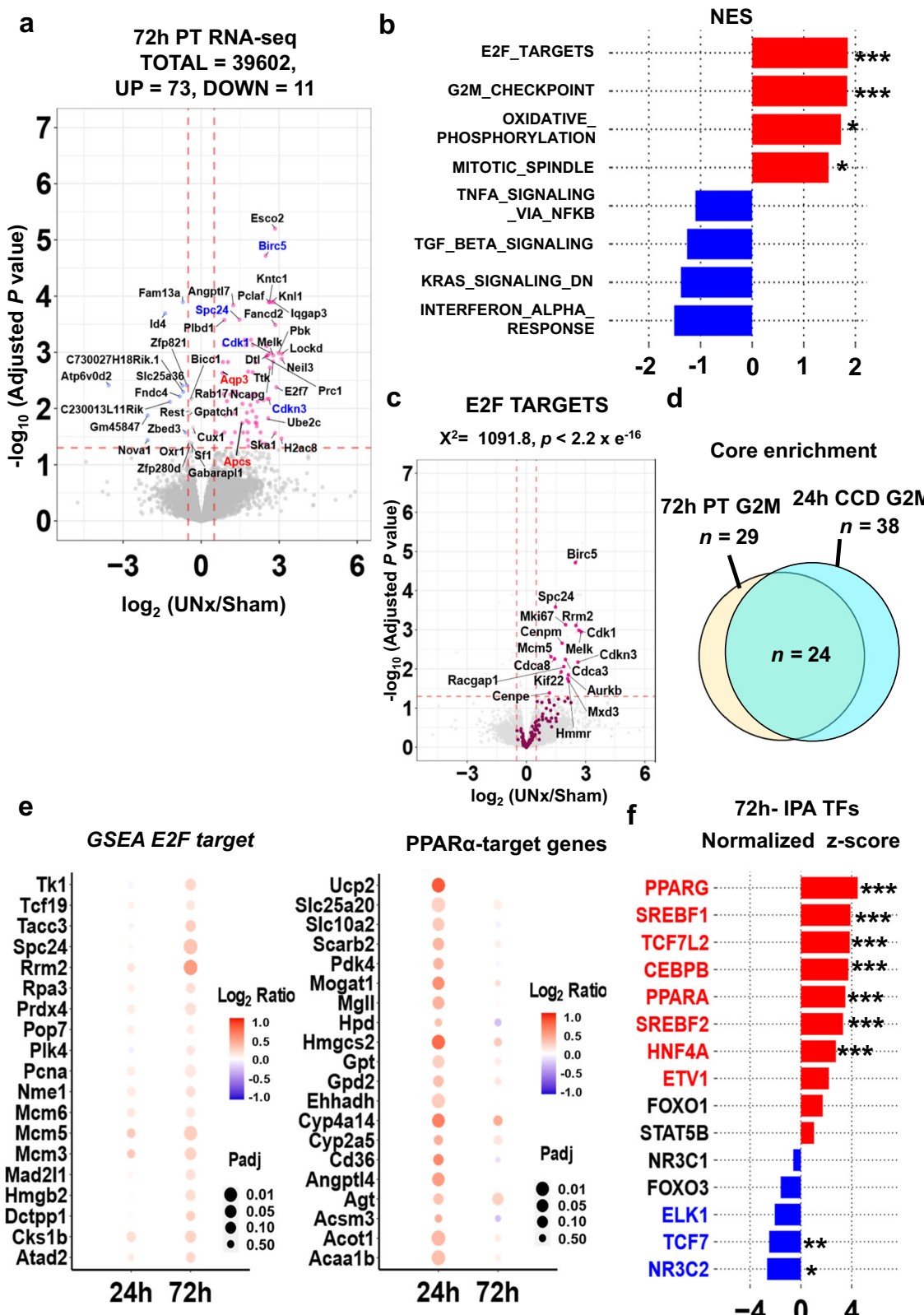

cholesterol and phospholipids including *Elovl1* and *Cct5*[41], both increased in unilateral nephrectomy. Whereas PPARα is mostly known for its ability to induce fatty acid oxidation, growing evidence points to a role of PPARα in regulation of lipogenesis dependent on members of the SREBP (sterol regulatory element binding protein) family[42,43]. Of particular note, cardiolipin synthase 1 (CRLS1), which is essential for the biosynthesis of the mitochondrial lipid cardiolipin[44], exhibited

significantly increased protein levels after UNx at 24 h. Interestingly, this was not matched by changes in *Crls1* mRNA, suggesting that the increased protein level arises via a post-transcriptional mechanism.

The process by which PPARα is upregulated after UNx is so far unclear. Hypothetically, it could be initiated by a change in abundance of endogenous ligands or via some unknown post-translational modification of the PPARα protein. PPARα endogenous ligands comprise a

**Fig. 6 | Single-tubule RNA-seq for microdissected S1 proximal tubules (PT-S1) from Sham and UNx mice transcripts at 72 h. a** Volcano plot. Magenta points, significantly increased expression. Blue points, significantly decreased expression. Significant differential expression was determined using thresholds of $p_{adj} < 0.05$ and $|\log_2(\text{UNx/Sham})| > 0.5$. Genes known to be regulated by PPARα in red font. Cell-cycle regulated genes highlighted in blue font. ($n = 5$ for both UNx and Sham). **b** Top-ranked Hallmark Pathway gene sets from Gene Set Enrichment Analysis (GSEA) in PT-S1 NES, normalized enrichment score. *$p < 0.05$, **$p < 0.01$, ***$p < 0.001$ (weighted Kolmogorov−Smirnov test; $p$ values are provided in Supplementary Table 2). **c** Volcano plot with magenta points indicating E2F target genes (GSEA), which were significantly enriched based on Chi-squared analysis. **d** Venn diagram of

genes annotated as "G2M_CHECKPOINT" that were enriched in proximal tubule RNA-seq dataset at 72-h and in cortical collecting duct RNA-seq datasets at 24-h. **e** Bubble plots showing differential changes between the 24 h and 72 h time points for E2F target genes and for PPARα target genes. Adjusted $p$ values (Benjamini and Hochberg method) are visualized in circle size, and $\log_2$ ratios are visualized in color. **f** Prediction of upstream regulatory transcription factors using Ingenuity Pathway Analysis (IPA). Predictions based on differentially expressed genes in UNx vs. Sham at 72 h. Genes with normalized z-score more than 2 are colored in red, and with normalized z-score less than −2 are colored in blue. *$p < 0.05$, **$p < 0.01$, ***$p < 0.001$ (right-tailed Fisher's Exact Test, $p$ values are provided in Supplementary Data 15).

wide variety of structurally diverse lipids, including unsaturated and saturated fatty acids, fatty acyl-CoA species, oxidized fatty acids, and oxidized phospholipids[30,45]. One major group of hypotheses posed at the beginning of this paper is related to the well described increase in single-nephron GFR (SNGFR) in response to nephron loss by UNx[9,11]. First, increases in luminal abundance of fatty acids due to increased SNGFR might lead to PPARα activation, similar to the effect of lipid-induced PPARα activation in liver[46]. In this study, we observed significant increases in total amounts of saturated and unsaturated fatty acids, as well as triglyceride and phospholipid in post-UNx kidney tissue. Future studies are needed to quantify PUFA-derived metabolites such as 19- and 20-hydroxy-eicosatetranoic acids (19- and 20-HETE) that are also known to activate PPARs[47].

Similar arguments could be made about increased amino acid filtration due to increased SNGFR and luminal uptake by proximal tubule cells, increasing intracellular amino acid concentrations, a known signal that mediates cell growth via the mTOR pathway[15,48]. A previous study showed that the concentrations of free methionine, alanine, tyrosine, valine and leucine in renal cortical tissue were increased two days after unilateral nephrectomy in rats[49]. Given the current consensus that essential amino acids (leucine in particular) are responsible for the amino acid stimulation of the mTOR pathway[50], mTOR activation implicated in this study could be induced by increased amino acid influx. Consistent with this hypothesis, GSEA analysis for RNA-seq at 24 h indicated that the mTORC1 pathway was activated. However, mTOR was indicated to be inhibited in the IPA-upstream analysis performed using kidney cortex proteomics data at 24 h. Further, mTOR phosphorylation at S2448, which is associated with kinase activation, was decreased at 72 h.

Another possibility is related to increased energy demand by individual proximal tubule cells caused by increased SNGFR after unilateral nephrectomy. Increased SNGFR is matched by an increase in solute and water reabsorption in the proximal tubule by a process called 'glomerulo-tubular balance'. Active transport in the proximal tubule is dependent on the Na-K-ATPase and increased transport increases ATP demand[51,52]. Metabolism of fatty acids and other lipids to match this demand is likely to alter the intracellular lipid profile in proximal tubule cells, potentially resulting in PPARα activation. One potential mediator of this action, AMPK, is seemingly ruled out by the findings that phosphorylation of its catalytic subunit (Prkaa1) at Thr172 and its regulatory subunit (Prkab1) at Ser108 are decreased in the remaining kidney after unilateral nephrectomy, pointing to inactivation rather than activation.

To investigate the mechanism of compensatory renal hypertrophy, we used an unbiased systems biology approach. Specifically, we observed changes in DNA accessibility, as well as mRNA and protein abundances at 24 h following unilateral nephrectomy. This strategy is based on the idea that in a causal chain of events, the earliest ones are the most likely to be related to the ultimate instigating signal, free of secondary responses. Such secondary responses are likely to be more prevalent in the 72-h data, resulting in apparently conflicting observations. Although PPARα was consistently identified as activated at both the 24-h and the 72-h time points, other mechanisms relevant to our

hypothesis showed differences. For example, one hypothesis we posited was increased AKT signaling induced by accumulation of growth factors[14]. AKT signaling was identified as activated at 24 h based on bioinformatic analysis and immunoblotting, while an upstream regulator of AKT, namely PDPK1[53] exhibited decreased Ser244 phosphorylation at 72 h, indicative of decreased activity. PDPK1 and AKT are key kinases activated in growth factor signaling in addition to the MAPK pathway. The phosphoproteomic data show decreased phosphorylation of ERK1 (Mapk3), indicating a decrease in activity, contrary to the increase that would be expected in response to growth factors such as IGF1[54]. In general, opposite changes in ERK1 and AKT give conflicting information about the possible role of growth factors in compensatory hypertrophy. These apparently conflicting observations are expected in complex signaling systems and will require focused studies to resolve.

In conclusion, we found that the lipid-regulated transcription factor PPARα plays a critical role in determination of kidney size and is likely to be a necessary mediator in compensatory kidney growth in response to unilateral nephrectomy. Compensatory hypertrophy occurs chiefly through increases in the size of proximal tubule cells, which account for most of the kidney cell mass. A contrasting response to unilateral nephrectomy is seen in the collecting duct, which displays cellular hyperplasia instead. This study also highlights the power of unbiased multi-omic approaches, which have advantages over purely reductionist approaches, to achieve understanding of complex signaling systems. This approach has empowered us to simultaneously investigate multiple potential mechanisms of renal tubular hypertrophy, which provide a foundation for future studies clarifying pathophysiological events associated with acute kidney disease (Supplementary Discussion 2), kidney transplantation, renal trauma, renal cancer and chronic kidney disease.

## Methods
### Animals
All animal experimental procedures were carried out in accordance with National Heart, Lung, and Blood Institute [NHLBI] animal protocol H-0047R6, approved by the National Heart, Lung and Blood Institute Animal Care and Use Committee. Pathogen-free, male, 6-to 8 week-old C57BL/6 mice (Taconic) were used. PPARα wild-type (PPARα$^{+/+}$) and PPARα-null (PPARα$^{-/-}$) mice (RRID:IMSR_JAX:008154), both are male C57BL/6 strain, were obtained from Dr. Frank J. Gonzalez in the Laboratory of Metabolism, Center for Cancer Research, National Cancer Institute, National Institutes of Health[33]. All mice were maintained on 12 h light dark cycles with the housing temperatures of 18−23 C with 40−60% humidity. The mice were assigned to two groups with sham operation or with unilateral nephrectomy (UNx). UNx was performed by the surgical excision method. Briefly, the mice were anesthetized and placed with the right side on a heating pad with medium heat, and a left flank skin incision (1–1.5 cm long) was made. The muscle layer was then incised by small scissors. The left kidney was removed through a left paramedian incision after ligation of the left renal artery, vein, and ureter. Sham-operated mice were anesthetized and only underwent skin and muscle incision without removal of any left kidney mass. In some experiments, the mice were given fenofibrate

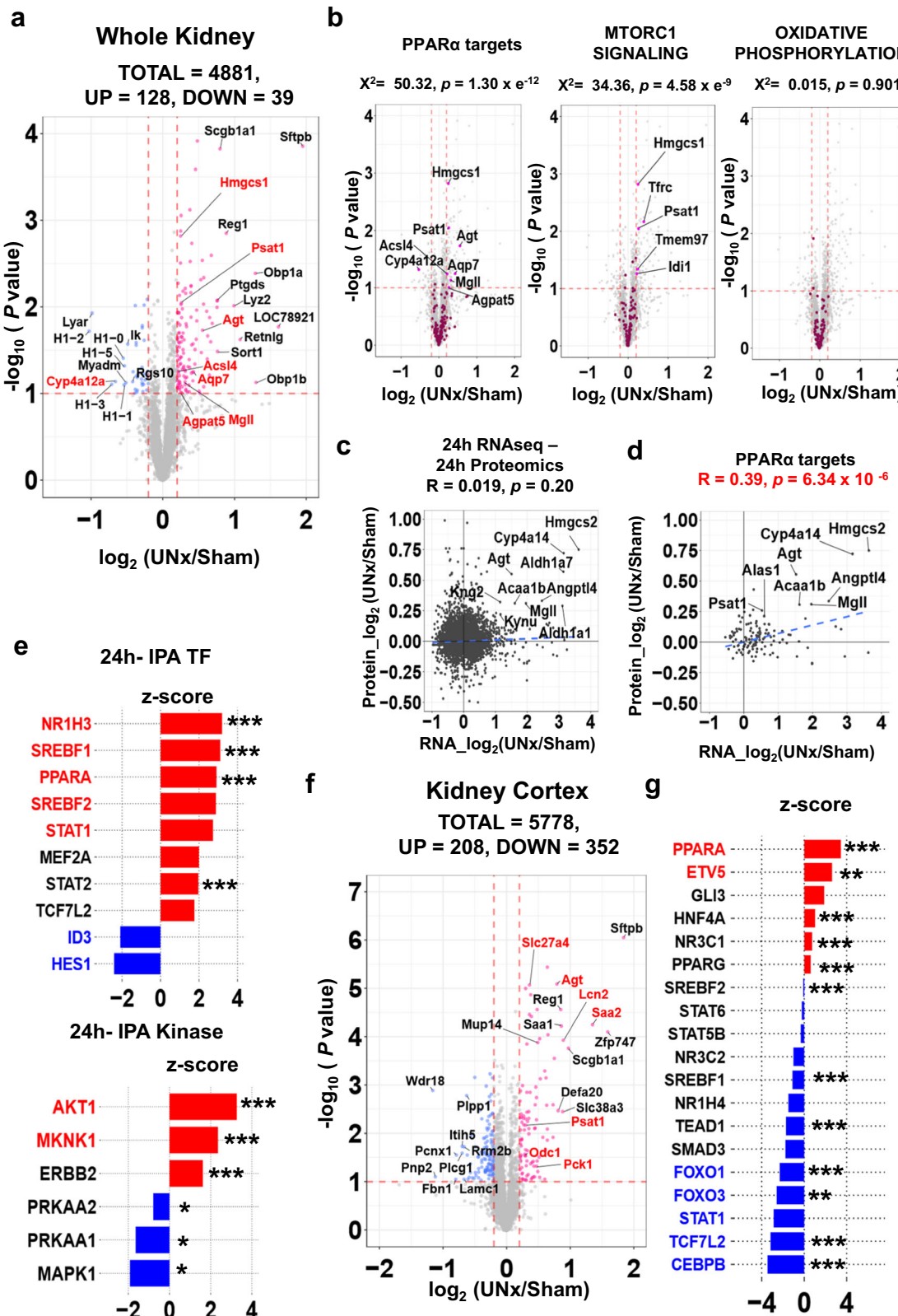

(2-[4-(4-Chlorobenzoyl)phenoxy]-2-methylpropanoic acid isopropyl ester, 50 mg/kg BW i.p. daily in 4% DMSO/PBS, Sigma # F6020)[55]. Control mice received only the vehicle.

**Microdissection of renal tubules from mice**

Mice were euthanized via cervical dislocation. The kidneys were perfused via the left ventricle with ice-cold Dulbecco's PBS (DPBS; Thermo

Scientific) to remove blood cells, followed by reperfusion with the dissection buffer (5 mM HEPES, 120 mM NaCl, 5 mM KCl, 2 mM calcium chloride $CaCl_2$, 2.5 mM disodium phosphate, 1.2 mM magnesium sulfate, 5.5 mM glucose, 5 mM Na acetate, pH 7.4) with 1 mg/ml collagenase B (Roche). We harvested the kidneys, obtained thin tissue slices along the cortical-medullary axis, and proceeded to digestion. The digestion was carried out in dissection buffer containing collagenase B

**Fig. 7 | Quantitative proteomics of whole kidneys from Sham and UNx at the 24-h timepoint.** Data from TMT-based quantitative proteomics using LC-MS/MS) of whole kidney from mice with either Sham ($n = 4$) or UNx ($n = 4$) surgery. **a** Volcano plot. Red, upregulated in UNx ($p < 0.1$ and $\log_2$ (UNx/Sham) > 0.2); blue, proteins downregulated in UNx ($p < 0.1$ and $\log_2$ (UNx/Sham) < −0.2). PPARα regulated proteins are highlighted in red font. **b** Volcano plots for the proteins known to be regulated by PPARα, and the proteins annotated as "MTORC1_SIGNALING" and "OXIDATIVE_PHOSPHORYLATION" in GSEA. (indicated by magenta points in respective plots). *p*-value represents the likelihood of enrichment of respective gene sets among regulated peaks using Chi-squared analysis. **c** $\log_2$ of the abundance ratio of all identified proteins plotted against $\log_2$ of the abundance ratio of transcripts identified in proximal tubule 24 h RNA-seq. Blue dashed line shows best-fit linear correlation (Pearson's product moment correlation coefficient using the stat_cor function in R). **d** $\log_2$ of the protein abundance ratio of PPARα target proteins plotted against $\log_2$ of the mRNA abundance ratio of PPARα target genes at the 24-h timepoint. Blue dashed line shows best-fit linear correlation (Pearson's product moment correlation coefficient using the stat_cor function in R). **e** Prediction of upstream regulatory transcription factors (top) and kinases (bottom) determined using Ingenuity Pathway Analysis (IPA). (Fisher's Exact Test, *p* values are provided in Supplementary Data 17). **f** Volcano plot for cortical samples. Sham ($n = 5$) or UNx ($n = 5$) surgery. Red dots, proteins upregulated in UNx ($p < 0.1$ and $\log_2$ (UNx/Sham) > 0.2); blue dots, proteins downregulated in UNx ($p < 0.1$ and $\log_2$ (UNx/Sham) < −0.2). PPARα-regulated proteins highlighted in red font. **g** Prediction of upstream regulatory transcription factors using Ingenuity Pathway Analysis (IPA) based on kidney cortex proteomics. $*p < 0.05$, $**p < 0.01$, $***p < 0.001$ (Fisher's Exact Test, *p* values are provided in Supplementary Data 19).

(1–2 mg/ml) at 37 °C with frequent agitation. We monitored the digestion until the optimal microdissectable condition was reached, typically 30 min. The microdissection was carried out under a Wild M8 dissection stereomicroscope equipped with on-stage cooling. The renal tubules were washed in dishes containing ice-cold DPBS by pipetting to remove contaminants before RNA extraction using a Direct-zol RNA MicroPrep kit (Zymo Research, Irvine, CA). Four to eight tubules were collected for each sample for a total length of 2–4 mm.

### Small-sample RNA-seq
mRNA collection and purification were performed as previously reported[18]. RNA was eluted in 10 μl of sterile water. cDNA was generated using the SMART-Seq V4 Ultra Low RNA Kit (Takara Bio, Mountain View, CA). After 14 cycles of library amplification, 1 ng of cDNA was "tagmented" and bar coded using a Nextera XT DNA Sample Preparation Kit (Illumina). The final libraries were purified using AmPure XP magnetic beads (Beckman Coulter, Indianapolis, IN) and quantified using a Qubit 2.0 Fluorometer (Thermo Fisher Scientific, Waltham, MA). Sample quantity and quality were assayed on an Agilent 2100 bioanalyzer. cDNA library concentrations were normalized, and samples were pooled and sequenced on an Illumina Novaseq 6000 platform using a 50 bp paired-end modality. Approximately 60–80 million reads were obtained from each library.

### RNA-seq initial processing
FastQC was used to evaluate sequence quality (software version 0.11.9) ((http://www.bioinformatics.babraham.ac.uk/projects/fastqc/).Adapter contamination was not significant, so read trimming was not performed. RNA-seq reads were indexed using STAR (2.7.6a) and aligned to the mouse reference genome from Ensembl (release 103) using STAR (2.7.6a)[56] with the matching genome annotation file (release 103). Default settings were used except for:–runThreadN $SLURM_CPUS_PER_TASK −outFilterMismatchNmax 3 −outSAMstrandField intronMotif −alignIntronMax 1000000 −alignMatesGapMax 1000000−outFilterIntronMotifs RemoveNoncanonicalUnannotated −outFilterMultimapNmax 1. Transcripts per million (TPM) and expected read counts were generated using RSEM (1.3.3)[57].Unless otherwise specified, the computational analyses were performed on the NIH Biowulf High-Performance Computing platform.

### Differential expression analysis
Raw expected counts (RSEM output) were used as input for this analysis. Genes whose sum of counts per million [CPM] values across all samples was fewer than 100 CPM were removed from downstream analysis. Differential expression was assessed using edgeR (v 3.40.2) and DESeq2 (v1.39.8) based on the previously described protocols of Anders et al.[58] and Love et al.[59], respectively, in concert with the user guide information provided for each package on the Bioconductor website. DESeq2[60] was used to apply a Benjamini-Hochberg false discovery rate (FDR) correction to the differential expression *p*-values to account for multiple hypothesis testing.

### Renal tubule ATAC-seq
ATAC-seq was performed using a modified method based on Corces et al.[61]. Three manually dissected proximal tubules (300–600 μm) were lysed and transposed simultaneously in 25 μl of transposition mix (0.1% NP40, 0.1% Tween-20, 0.01% digitonin and 5% Tn5 enzyme (Illumina # 15027865) in Tagment DNA Buffer (FC-121–1030; Illumina). The transposition reaction was incubated at 37 °C for 30 min in a thermomixer with shaking at 1000 rpm. After tagmentation, the reactions were stopped with addition of 80 μl of water, 3 μl of 0.5% EDTA, 1.5 μl of 20% SDS and 5 μl of proteinase K. The transposed DNA fragments were purified by DNA Purification Kits (Zymo Research, # D4014), and amplified using PCR. The final libraries were purified using AmPure XP magnetic beads (Beckman Coulter, Indianapolis, IN). Mitochondrial DNA were depleted from the purified libraries using a CRISPR/Cas9-based method[62]. Final library was quantified using a Qubit 2.0 Fluorometer (Thermo Fisher Scientific, Waltham, MA). Purified DNA libraries were sequenced on an Illumina Novaseq platform.

### ATAC-seq initial processing
Adapter sequences were trimmed for both forward and reverse reads using cutadapt (version 3.4, parameters: minimum length = 36,–q = 30). Read quality was assessed using fastQC (v 0.11.9). Trimmed sequences were aligned to the mouse reference genome mm10 using bowtie2 (v 2.4.5, parameters: X = 2000,–no-mixed,–no-discordant)[63]. The resulting SAM files were converted to a binary format (BAM), sorted by queryname, and indexed using samtools (v 1.14, function used; samtools view, samtools sort and samtools index)[64]. Reads that mapped to the mitochondrial genome were removed using samtools (function used; samtools idxstats). Identically mapping read duplicates were marked using Picard (v 2.25.7) MarkDuplicates (Broad Institute. Picard toolkit. Broad Institute, GitHub repository (2019)), and removed using samtools (v. 1.14, parameters: -F 1804, -f 2, -q = 30). BAM files were converted to BED format using BedTools (v2.30.0)[53]. Narrow open chromatin peaks were identified using MACS2 with parameter −nomodel−shift -75 −extsize 150[65]. Reads mapping to ENCODE blacklist regions[66] were discarded using BedTools (v 2.30.0) (These are empirically identified genomic regions that produce artifactual high signal in functional genomic experiments). For visualization, bigwig files were generated using bamCoverage. Read distribution was visualized on the UCSC genome browser. As ATAC-seq-specific quality control, fragment size distribution were evaluated by the *bamQC* function under the R package *ATACseqQC* (v 1.22.0)[67].

### Differential accessibility analysis
Differentially accessible regions (DAR) between Sham ($n = 3$) and UNx ($n = 3$) were identified using R package DiffBind (v 3.8.4) (R. Stark, G. Brown, DiffBind: Differential Binding Analysis of ChIP-Seq Peak Data, Available from: https://bioconductor.org/packages/release/bioc/vignettes/DiffBind/inst/doc/DiffBind.pdf), with the following statistical cutoff: FDR < 0.01. Among the 125,973 open chromatin regions identified among all samples, 4223 were identified as differentially

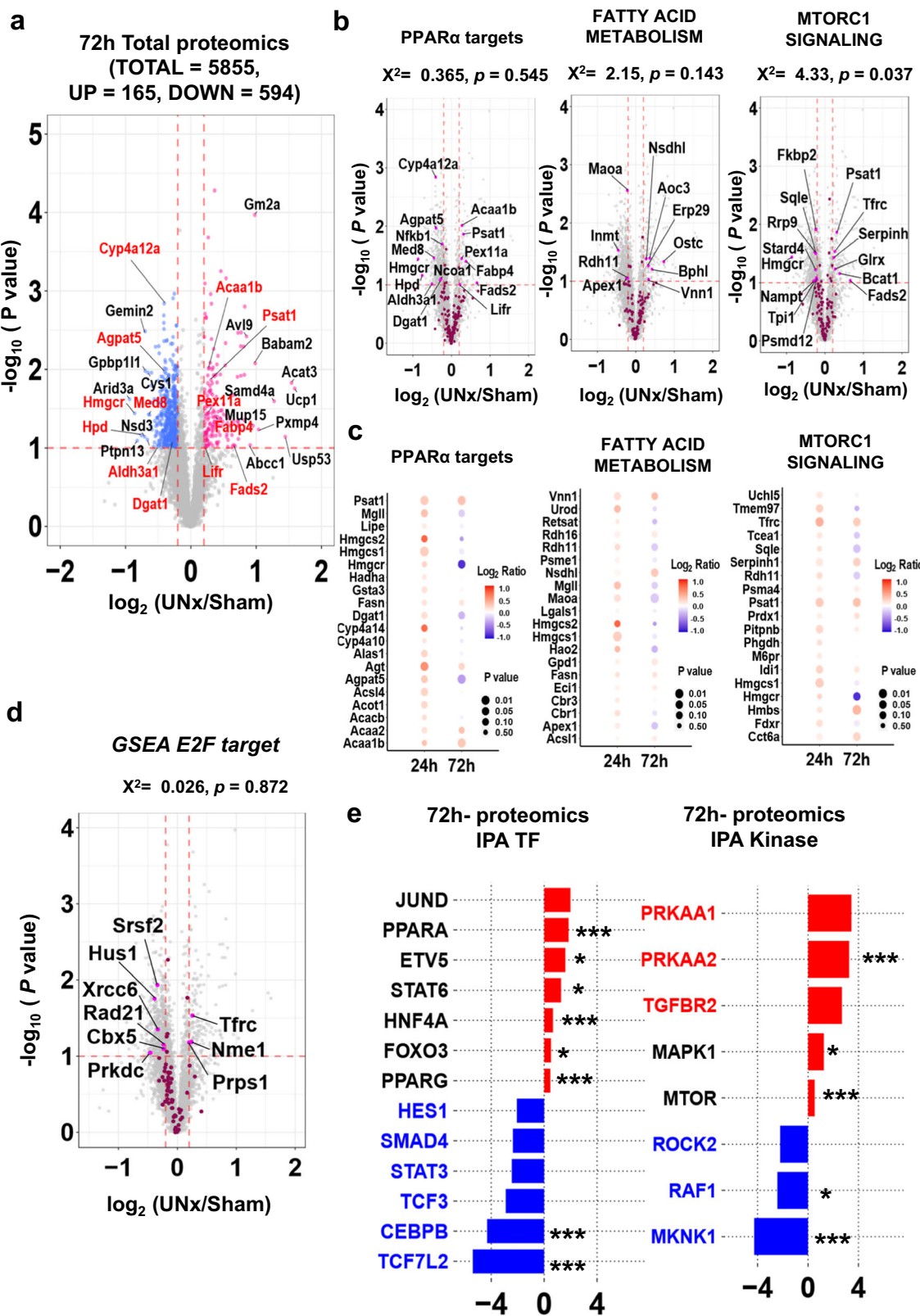

accessible. As a downstream analysis, the footprint analysis was done by the *factorFootprints* function under the R package *ATACseqQC* (v 1.22.0)[67].

**Gene-set enrichment analysis (GSEA)**
GSEA (http://software.broadinstitute.org/gsea/index.jsp) was used to estimate the enriched gene ontology (GO) terms[68]. We used mouse gene sets database downloaded from Bader Lab (http://download.baderlab.org/EM_Genesets/) that contained all mouse GO terms as gene set file input to GSEA. GSEA pre-ranked analysis (GseaPreranked) was performed using default settings except for "Collapse dataset to gene symbols" set to "No-Collapse." Prior to analysis, a ranked list was calculated with each gene assigned a score based on the FDR and the $\log_2$ (UNx/Sham).

**Fig. 8 | Quantitative proteomics of whole kidneys from Sham and UNx at the 72 h timepoint. a** Volcano plot for UNx vs. Sham. TMT-based quantitative proteomics LC-MS/MS of whole kidneys from mice with either Sham ($n = 4$) or UNx ($n = 4$) surgery. Red dots, upregulated proteins upregulated in UNx ($p < 0.1$ and $\log_2$ (UNx/Sham) > 0.2); blue dots, downregulated proteins in UNx ($p < 0.1$ and $\log_2$ (UNx/Sham) < −0.2). PPARα regulated proteins in red font. **b** Volcano plots highlighting proteins known to be regulated by PPARα, and proteins with roles in "FATTY ACID METABOLISM" and "MTORC1_SIGNALING" in GSEA indicated by magenta points. *p*-values associated with enrichment of respective gene sets (Chi-squared analysis). **c** Bubble plots showing differential changes of proteins between the 24 h and 72 h time points for the proteins regulated by PPARα, and for the proteins annotated as "FATTY ACID METABOLISM" and "MTORC1_SIGNALING" in GSEA. *P* values (unpaired, two-tailed t-test) are visualized by circle size, and $\log_2$ ratios are visualized by color. **d** Volcano plot for the proteins annotated as "E2F TARGET" in GSEA. Magenta points, known E2F targets. Significant differential abundance was determined using thresholds of $p < 0.1$ (unpaired, two-tailed *T*-test) and $|\log_2$ (UNx/Sham)| > 0.2. *p*-value represents the likelihood of E2F target protein enrichment among regulated peaks using Chi-squared analysis. **e** Prediction of upstream regulatory transcription factors (left) and kinases (right) determined using Ingenuity Pathway Analysis (IPA) at 72 h. *$p < 0.05$, **$p < 0.01$, ***$p < 0.001$ (Fisher's Exact Test, *p* values are provided in Supplementary Data 21).

## Pathway analyses

Ingenuity Pathway Analyses (IPA, (https://www.qiagenbioinformatics.com/products/ingenuity-pathway-analysis/, IPA, Qiagen) was used for identifying upstream regulatory molecules/networks listed in Supplementary Data 4 using RNA-seq and Proteomics dataset. The upstream regulator analysis tool is a novel function in IPA which can, by analyzing linkage to DEGs through coordinated expression, identify potential upstream regulators including transcription factors (TFs) and any gene or small molecule that has been observed experimentally to affect gene expression[69].

## Target gene-set analysis

Target genes for TFs including SREBF1, SREBF2, CREM, SMAD4, NR1H3, JUN, JUND, STAT1, STAT3, STAT6, E2F4 and NR3C1 were curated from either mammalian ChIP-seq datasets (ENCODE Transcription Factor Targets dataset[70,71] or CHEA Transcription factor targets dataset[72]. Other curated genes lists included PPARα[73–75], HNF4α[76], NR1H2[77], and NR1H4[77,78]. Target gene sets for Id2, Ybx1, Etv1 and Smad2 were not available (Supplementary Table 1). For ATAC-seq, all peaks are used without any filtering condition. For RNA-seq, genes whose TPM are less than 1, are filtered out from analysis. Transcriptional activities of curated transcription factors were estimated by comparing the distributions of $\log_2$ ratios (UNx/Sham) of peak concentrations (in ATAC-seq) and TPM values (in RNA-seq) for TF-target gene sets. Statistical significance was evaluated using an un-paired, two-tailed Student's *t*-test. $p < 0.05$ was considered statistically significant. Data are presented as mean $\pm 1.96 \times$SD (95% confidence interval).

## Quantitative PCR

Total RNA was extracted from mouse kidneys and livers by the Direct-zol™ RNA Purification Kits (Zymo Research). Purified RNA was reverse transcribed using the SuperScript™ IV First-Strand Synthesis System (Thermo Fisher Scientific). cDNAs from the mouse kidney and liver ($n = 3$-4 mice for each) were used for quantitative PCR (qPCR). qPCR assays were performed using FastStart Universal SYBR Green Master mix (Sigma) according to the manufacturer's protocol with minor modifications. In the 96-well plate (Applied Biosystems), 15-μl reactions (10 ng cDNA) were performed on a LightCycler® 96 System (Roche). The change in the gene expression was calculated using $2^{(-\Delta\Delta Ct)}$ method. The amounts of mRNA were normalized to β-actin, and were calculated using the comparative CT method. Sequences for the qRT-PCR primers employed are described in Supplementary Table 4.

## Quantitative immunocytochemistry in microdissected tubules

Determination of the numbers of each cell type per unit length in microdissected PTs and CCDs from UNx and Sham mice was carried out using immunocytochemistry employing antibodies recognizing cell type specific markers, based on Purkerson et al.[79]. The primary antibodies used were rabbit anti-AQP1 (LL266, in house, 1:100), mouse anti V-ATPase B1/B2 (F-6, sc-55544, Santa Cruz Biotechnology, Santa Cruz, CA, 1:100), anti-chicken AQP2 (CC 265, in house, 1:1000) and Alexa Fluor 568 phalloidin (A12380, Invitrogen, 1:400). The secondary antibodies were Alexa Fluor 488 goat anti-rabbit, Alexa Fluor 488 goat anti-chicken, Alexa Fluor 568 goat anti-chicken and Alexa Fluor M-594 goat anti-mouse IgG. (A11034, A11039, A11041 and A11032, Invitrogen) each at 1:400 dilution. Cell nuclei were labelled with DAPI. Confocal fluorescence images were recorded with a Zeiss LSM780 confocal microscope using a 20× objective lens by Z-stack scanning. 3D images are reconstructed using z-stack files, and cell counting was performed on three-dimensional reconstructed tubule images using IMARIS Scientific Image Processing & Analysis software (v7.7.1, Bitplane, Zurich, Switzerland). Counting was automated using IMARIS "spot analysis" for nuclei. Tubule volume was calculated using IMARIS "surface analysis".

## Histologic analysis and immunocytochemistry

Mice underwent cervical dislocation and were perfused with ice-cold DPBS followed by 4% paraformaldehyde in DPBS. Whole kidneys and liver were then maintained for 2 h in 4% paraformaldehyde before transferring to 20% sucrose at 4 °C overnight. Kidney and liver samples were embedded in O.C.T. compound (Sysmex). Cryosections (6 μm thick) were cut for histological analysis with H-E staining. For immunofluorescence staining, frozen sections were thawed at room temperature for 10–20 min and rehydrated in PBS for 10 min. After blocking for 30 min with 1% BSA and 0.2% gelatin, primary antibodies were applied overnight at 4 °C. The primary antibodies used were chicken anti-AQP2 (CC 265, in house, 1:1000) and rabbit anti Ki-67 (Abcam # 16667, RRID:AB_302459, 1:100). Sections were washed three times for 5 min in PBS. The secondary antibody incubation was carried out for 1 h at room temperature. The secondary antibodies used were Alexa Fluor R-568 goat anti-rabbit, Alexa Fluor C-488 goat anti-chicken, each at 1:200 dilution. Stains were analyzed using a Zeiss LSM780 confocal microscope using ZENBlue software (v. 3.7, Zeiss).

## Immunoblotting of kidney tissue

For immunoblotting experiments, the mice were euthanized by decapitation at 24 h and 72 h after surgery ($n = 3$–4 in each group). The whole kidney was homogenized on ice (CK-100 Tissue Homogenizer, 15sX4) in isolation solution (250 mM sucrose, 10 nM Triethanolamine, pH = 7.6) with HALT protease/phosphatase inhibitor mixture (Thermo Scientific). After determining protein concentrations (Pierce BCA Protein Assay Kit), samples were lysed in Laemmli buffer. Immunoblotting was performed, using 12% polyacrylamide gels (BioRad) using 20 μg protein per lane. After transfer to a nitrocellulose membrane, the membrane was probed overnight with anti-AMPK-alpha (Thr172) antibody (rabbit, 1:1000, Cell signaling # 2535, RRID:AB_331250), anti-AMPK-α antibody (rabbit, 1:1000, Cell signaling # 2532, RRID:AB_330331), anti-AKT antibody (rabbit, 1:1000, Cell signaling # 9272, RRID:AB_329827), anti-AKT (Ser473) antibody (rabbit, 1:1000, Cell signaling # 9271, RRID:AB_329825), anti-PPARα antibody (rabbit, 1:1000, NOVUS Biologicals # NB600-636), anti-NaPi-2 antibody (rabbit, 1:1000, in house #L697) or anti-AQP2 (rabbit, 1:1000, in house #K5007). After incubation with goat anti-rabbit IRDye 680 secondary antibodies (LI-COR) for 1 hr, blots were imaged on an Odyssey CLx

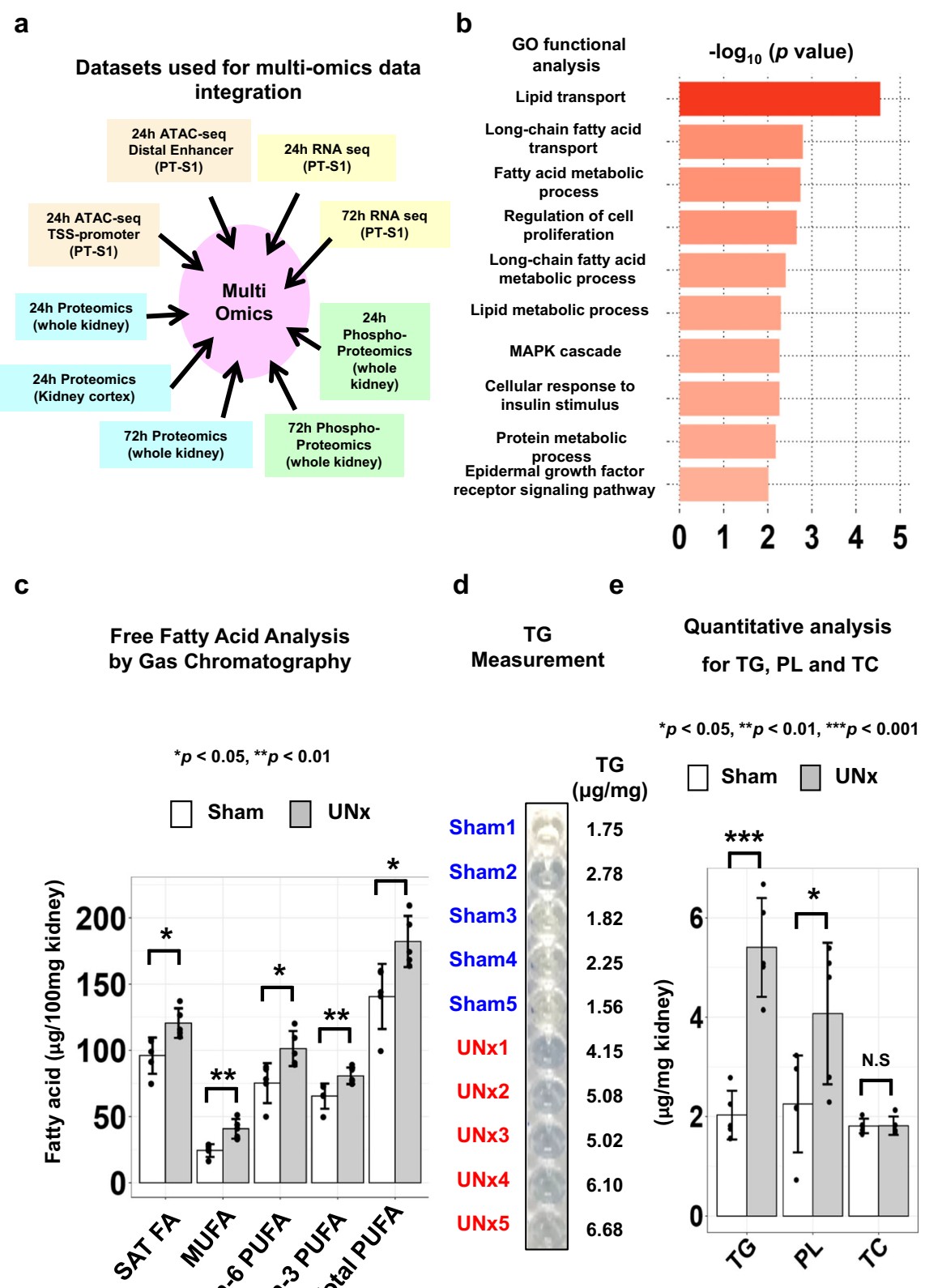

In brief, cytoplasmic extraction reagents I and II were added to a lysate to disrupt cell membranes, releasing cytoplasmic contents. After recovering the intact nuclei from the cytoplasmic extract by centrifugation, the nuclei were lysed with a nuclear extraction reagent to yield the nuclear extract. Immunoblot analysis was used to assess the adequacy of nuclear purification by measuring Lamin A/C (rabbit, 1:1000, Cell signaling # 2032, a nuclear protein).

Imaging System (LI-COR Biosciences) and band densities was quantified using associated software.

**Preparation of nuclear fractions**
The whole kidney was homogenized, and nuclear fraction was extracted by NE-PER nuclear and cytoplasmic extraction reagents (Thermo Scientific, # 78833), following the manufacturer's instruction.

**Fig. 9 | Multi-omics data integration and lipid analysis. a** Datasets used for multi-omics data integration using DAVID. **b** GO functional enrichment in biological process using output from data integration. The selected 10 significantly enriched GO biological process terms relevant to original hypothesis proposed in Fig. 2 and Supplementary Data 4. ($p < 0.05$, Fisher's Exact test, $p$ values are provided in Supplementary Data 25). GO gene ontology. **c** Concentrations of total saturated fatty acids (SAT FA, $p = 0.013$), monounsaturated fatty acids (MUFA, $p = 0.007$), n-6 and n-3 poly unsaturated fatty acid (PUFA, $p = 0.015$, $p = 0.008$), and total PUFA ($p = 0.012$) in Sham ($n = 5$) versus UNx ($n = 5$), analyzed by gas chromatography in kidney tissue. Data are presented as mean ± SD. *$p < 0.05$, **$p < 0.01$ (unpaired, two-tailed $T$-test). **d** Representative image of colorimetric assay for triglycerides comparing sham and UNx samples. Uncropped image is provided as a Source Data file. **e** Concentration of triglycerides (TG, $p = 0.00013$), phospholipid (PL, $p = 0.046$) and total cholesterol (TC, $p = 0.976$) by colorimetric quantitative analysis in kidney tissue from UNx ($n = 5$) vs. Sham ($n = 5$). Data are presented as mean ± SD. *$p < 0.05$, **$p < 0.01$, ***$p < 0.001$ (unpaired, two-tailed $T$-test).

## Quantification and statistical analysis

The band intensities of the Western blots were quantified using Image J software (National Institutes of Health, Bethesda, MD). Statistical significance was evaluated using an un-paired, two-tailed Student's $t$-test. $p < 0.05$ was considered statistically significant. Data are presented as mean ± standard deviation (SD) or mean ± standard error of the mean (SEM) or mean ± 1.96×SD (95% confidence interval). All statistical methods used are summarized in Supplementary Method. Asterisks denote corresponding statistical significance *$p < 0.05$, **$p < 0.01$ and ***$p < 0.001$. Statistical analyses were carried out with SPSS 17.0 statistical software (SPSS, Inc., Chicago, IL, USA).

## Sample preparation: quantitative protein mass spectrometry

A whole kidney or kidney cortex from each mouse was collected and homogenized by polytron based homogenizer for 15 s three times in 1 ml of cold 100 mM TEAB buffer with 1X HALT protease and phosphatase inhibitor cocktail (Thermo Scientific, # 1861280). Lysates were centrifuged at 700 x G for 5 mins at 4 C. Supernatant were used for BCA Protein Assay. 700 µg (for total and phosphor-proteomics) of proteins from supernatant were transferred to a new tube and adjusted to a final volume of 100 ul with 100 mM TEAB buffer. Samples were reduced by incubation with 5 µl of the 500 mM DTT for 1 h, followed by alkylation with 5 µl of the 375 mM of IAA (iodoacetamide) both at room temperature. For protein precipitation, 600 µl of pre-chilled acetones were added and incubated at −20 °C overnight. The precipitated proteins were harvested by centrifugation at $8000 \times g$ for 10 min at 4 °C. After removal of acetone, the precipitated protein samples were digested with Trypsin/LysC (Promega) (1:50 wt/wt.) in 100 mM TEAB at 37 °C for 18 h. The digested peptides were quantified using Pierce Quantitative Colorimetric Peptide Assay (Thermo Fisher Scientific), and stored at −80 °C until the TMT labeling.

## TMT labeling

Equal amounts (400 µg) of peptides from each sample were taken and the volume was adjusted to 100 µl of 100 mM TEAB, then labeled with TMT Isobaric Mass Tag (TMT11Plex, Thermo Fisher Scientific) following the manufacturer's instructions. After labeling, all samples were pooled and desalted using hydrophilic-lipophilic-balanced extraction cartridges (Oasis), then fractionated using high pH reverse-phase chromatography (Agilent 1200 HPLC System). The fractionated samples were dried in a SpeedVac (Labconco) and stored in at −80 °C.

## Phosphopeptide enrichment

From each fraction, 5% was collected in a separated tube for "total" proteomics and the remaining 95% was further enriched for "phospho" proteomics. To enhance phosphopeptide identification, we followed the Sequential Enrichment from Metal Oxide Affinity Chromatography protocol (SMOAC) from Thermo Fisher Scientific for the phosphopeptide enrichment. In brief, pooled TMT-labeled peptides were first processed with the High-Selected $TiO_2$ kit (Thermo Fisher Scientific), and then the flow through was subsequently subjected to the High-Selected Fe-NTA kit (Thermo Fisher Scientific) per manufacturer's instructions. The eluates from both enrichments were combined, dried and stored at −80 °C until LC-MS/MS analysis.

## Liquid chromatography-tandem mass spectrometry (LC-MS/MS)

Total peptides and phospho-enriched peptides were reconstituted with 0.1% formic acid in LC-MS grade water (J.T. Baker) and analyzed using a Dionex UltiMate 3000 nano LC system connected to an Orbitrap Fusion Lumos mass spectrometer equipped with an EASY-Spray ion source (Thermo Fisher Scientific). The peptides were fractionated with a reversed-phase EASY-Spray PepMap column (C18, 75 µm × 50 cm) using a linear gradient of 4% to 32% acetonitrile in 0.1% formic acid (120 min at 0.3 µL/min). The default MS2 workflow was selected on the mass spectrometer for TMT quantification.

## Mass spectrometry data processing and analysis

The raw mass spectra were searched against the mouse UniProt reference proteome (UP000002494_10116.fasta, downloaded in August 2020) using MaxQuant 1.6.17.0, and lot-specific TMT isotopic impurity correction factors were used as recommended in the TMT product data sheets. "Trypsin/P" was set as the digestion enzyme with up to two missed cleavages allowed. Carbamidomethylation of cysteine (C) was configured as a fixed modification. Variable modifications included phosphorylation of serine, threonine and tyrosine (S, T, Y), oxidation of methionine (M). The FDR was limited to 1% using the target-decoy algorithm. Other parameters were kept as the defaults. Results are reported as MS2 reporter ion intensity ratios between UNx samples and Sham controls. The proteomics data are deposited to the ProteomeXchange Consortium via the PRIDE partner repository with the data identifiers the data identifiers PXD036395 (whole kidney) and PXD039697 (kidney cortex).

## Multi-omics data integration analysis

All $\log_2$ values from each nine omics datasets (ATAC-seq for TSS-promoter regions, ATAC-seq for Intergenic regions, RNA seq for proximal tubule S1 segment at 24 h, RNA seq for proximal tubule S1 segment at 72 h, proteomics for whole kidney at 24 h, proteomics for kidney cortex at 24 h, proteomics for whole kidney at 72 h, phosphoproteomics for whole kidney at 24 h and phosphoproteomics for whole kidney at 72 h) were selected out, and were normalized into z scores to determine differential enrichment or expression status for each ATAC-peak, gene or protein ($z > 1.96$, or $z < −1.96$) between UNx and sham treatments. Differentially enriched/expressed factors were assigned the logical score = 1, while unchanged factors were assigned a logical score = 0. Finally, all logical scores were summed up, and genes (proteins) with sum of logical score more than 3 were included in the DAVID analysis to identify enriched Gene Ontology Biological Processes.

## Fatty acid analysis

Total lipid from kidney tissue homogenates was extracted using a one-step method developed by Lepage and Roy[80]. In brief, lipids resuspended in methanol/hexane (4:1, v/v) with 50 µg/ml of butylated hydroxytoluene as antioxidant and tricosanoic acid (C23:0) as internal standard were subjected to transesterification at 100 °C for 1 h with acetyl chloride (Sigma Aldrich, St. Louis, MO, USA) in nitrogen atmosphere. The resulting fatty acid methyl esters (FAME) were separated and analyzed using gas chromatography on a Shimadzu GC2030 (Shimadzu Scientific Instruments, Columbia, MD) equipped

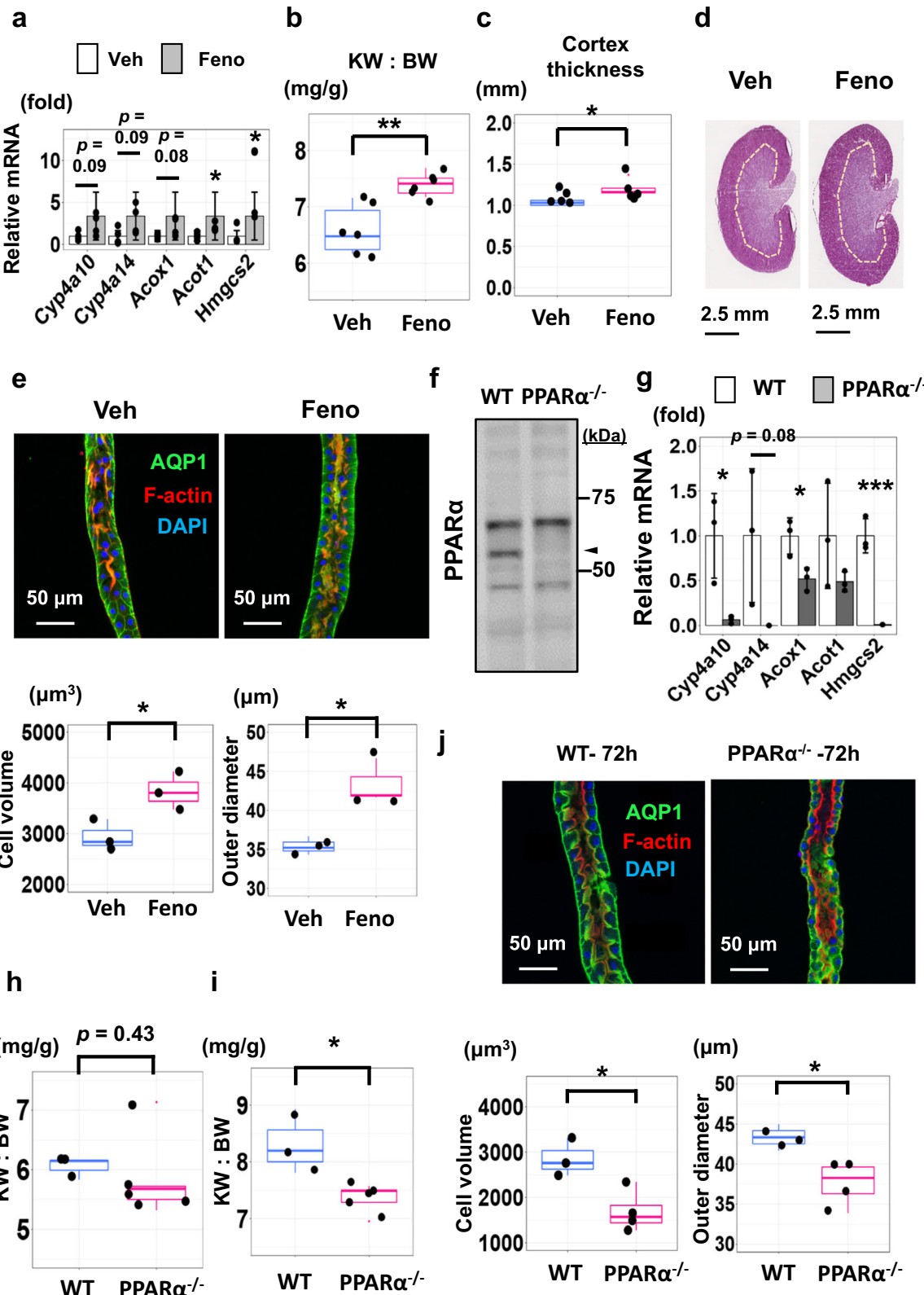

with an flame ionization detector and a capillary SLB®-IL111 column (30 m × 0.2 μm, φ0.25 mm; Sigma-Aldrich). Helium was used as carrier gas at a constant flow of 1.5 mL/min. The FAMEs were identified by comparing their retention time with purified FAME standards (Nu-Chek Prep, Elysian, MN). Tissue fatty acid concentration was calculated by using internal standard peak area, and the content of fatty acid was presented as μg per 100 mg of kidney wet weight.

**Quantification of kidney lipid content**

Total lipids were extracted from tissues following the Bligh and Dyer method[81]. A portion of kidney tissues (approximately 60 mg) were extracted with a mixture of methanol/chloroform/water (2:2:1.8, v/v/v). After the chloroform was evaporated under nitrogen gas, dried lipid was dissolved with ethanol containing 1% (v/v) Triton X-100. The total triglyceride, phospholipid, and cholesterol levels were measured using

**Fig. 10 | PPARα regulates cell size in the renal proximal tubule. a** Expression of PPARα target genes in the kidney determined using qRT-PCR. Fenofibrate (Feno, grey) vs vehicle (Veh, white). Mean ± SD ($n = 4$). **b** KW:BW ratios in mice treated with fenofibrate (Feno) or vehicle (Veh) for 14 days. ($n = 6$ for each group). ** $p = 0.0018$, unpaired, two-sided student $t$-test. **c** Cortical thickness of the kidneys in mice treated with fenofibrate versus vehicle ($n = 5$ for vehicle, $n = 6$ for fenofibrate). * $p = 0.0013$. **d** Hematoxylin and eosin stained kidneys from mice treated with vehicle (Veh) or fenofibrate (Feno) for 14 days. **e** Upper; representative confocal fluorescence image of a microdissected proximal tubules (S2 segment, PT). Lower; cell volume (left) and tubular outer diameter (right) with fenofibrate versus vehicle ($n = 3$ for each group). * $p < 0.05$ (Cell volume: $p = 0.034$, Outer diameter: $p = 0.010$). **f** Western blot for PPARα (nuclear protein fractions from wild-type versus PPARα$^{-/-}$ kidneys). Arrowhead, expected molecular weight. Data are representative of

biological replicates ($n = 7$ for wild-type, $n = 9$ for PPARα$^{-/-}$). Source data are provided as a Source Data file. **g** Expressions of PPARα target genes in wild-type (WT, white) or PPARα$^{-/-}$ (grey) mice (qRT-PCR). Mean ± SD ($n = 3$). **h** KW:BW in WT and PPARα$^{-/-}$ mice for the resected left kidney at UNx surgery; $n = 3$ for WT, $n = 5$ for PPARα$^{-/-}$. ($p = 0.43$, unpaired, two-sided student $t$-test). **i** KW:BW ratios in WT and PPARα$^{-/-}$ mice for the remnant kidney 3 days after UNx surgery. $n = 3$ for WT, $n = 5$ for PPARα$^{-/-}$. * $p = 0.015$. (unpaired, two-sided student $t$-test). **j** Upper; representative confocal fluorescence images of a microdissected proximal tubule (S2 segment) from WT and PPARα$^{-/-}$ mice 3 days after UNx surgery. Lower; cell volume (left), tubular outer diameter (right) in PPARα$^{-/-}$ versus WT mice. $n = 3$ for WT, $n = 4$ for PPARα$^{-/-}$. * $p < 0.05$. (Cell volume: $p = 0.019$, Outer diameter: $p = 0.028$, unpaired, two-sided student $T$-test). Box-and-whisker plots represent median and 25th and 75th percentiles-interquartile range.

enzymatic colorimetric assay (FUJIFILM WAKO Diagnostics U.S.A. Corporation) according to the manufacturer's instructions. The kidney lipids are expressed as μg per mg of kidney wet weight.

### Measurement of cell area sizes in liver

H-E stained liver sections was used for the quantification of cell area in the liver from mice treated either with vehicle or fenofibrate. All slides were scanned using the NDP Nanozoomer HT from Hamamatsu Photonics. The NDP Nano-zoomer produces virtual images of full tissue scans which have been analyzed visually as well as by automatic image processing algorithms. The full tissue sections allow large scale histological evaluations with high precision across the complete section. Liver cell area analysis was done using Hamamatsu NDP viewer. Twenty stained liver cells were selected randomly for cell area calculation for each biological replicates ($n = 4$ for vehicle, $n = 6$ for fenofibrate).

### Statistics and reproducibility

Statistical methods for each data element are described in figure legends. Replicate numbers (n) are biological replicates and not technical replicates in all cases. Source data are provided as a *Source Data* file.

### Reporting summary

Further information on research design is available in the Nature Portfolio Reporting Summary linked to this article.

### Data availability

Raw fastq files and raw count information from the RNA-seq analysis and ATAC-seq analysis were deposited on the GEO (GSE211021, GSE211022): and RNA-seq and ATAC-seq data can be browsed or downloaded via a Shiny-based web page at https://esbl.nhlbi.nih.gov/UNx/. Alternatively, ATAC-seq data is viewable at https://esbl.nhlbi.nih.gov/IGV_mo/ in IGV web browser. The proteomics data are deposited to the ProteomeXchange Consortium (https://proteomecentral.proteomexchange.org/cgi/GetDataset) via the PRIDE partner repository (https://www.ebi.ac.uk/pride/) with the data identifiers PXD036395 and PXD039697. To allow users facile access to the curated proteomics data, we have set up publicly accessible web resources at https://esbl.nhlbi.nih.gov/Databases/UnX-proteome/index.html for bulk kidney proteomics, and https://esbl.nhlbi.nih.gov/Databases/UnX-Phospho/72hlog.html for bulk kidney phosphoproteomics. Source data are provided as a Source Data file. Supplementary Data 1 through 31 files available at https://esbl.nhlbi.nih.gov/Databases/UNx-Supp/. CHEA Transcription factor targets dataset is available at https://maayanlab.cloud/Harmonizome/dataset/CHEA+Transcription+Factor+Targets. Source data are provided with this paper.

### Code availability

This paper did not employ author-derived programs. Access information to software used is given at https://esbl.nhlbi.nih.gov/Databases/UNx-Supp/.

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

## Acknowledgements

The work was funded by the Division of Intramural Research, National Heart, Lung, and Blood Institute (NHLBI project ZIA-HL001285 and ZIA-HL006129, M.A.K.). Some of the results were presented at the American Society of Nephrology Annual Meeting 2021 (Virtual). The authors thank Dr. Zu-Xi Yu, NHLBI/NIH, for expert technical assistance of histologic analysis, Dr. Frank J. Gonzalez and Dr. Shogo Takahashi, NCI/NIH, for giving PPARα knock out mice, Dr. Yanling Yang, NHLBI/NIH, for her technical support in the proteomics analysis, all of the staff of the NHLBI DNA Sequencing Core NHLBI/NIH, and Dr. Christian Combs and Dr. Daniela Malide, NHLBI/NIH (NHLBI Light Microscopy Core) and all of the staff in the NHLBI Animal Surgery Core. The authors thank Dr. Keita Saeki, NICHD/NIH, for expert technical assistance of processing files for sequencing data. Additionally, the authors thank Yolanda L. Jones and Brigit S. Sullivan, MLS, NIH Library, for editing assistance.

## Author contributions

H.K. and M.A.K. designed the experiments, H.K., C.L.C., C.R.Y., Z.H.Y., J.K. conducted the experiments, and H.K., K.L., L.C., H.J.J., B.C., Z.H.Y., E.P., J.K., A.T.R. and M.A.K. analyzed the data. C.L.C., L.C., K.L., H.J.J., A.T.R. and M.A.K. provided reagents and technique support. H.K., B.C. and M.A.K. wrote the manuscript. H.K., C.L.C., C.R.Y., H.J.J., E.P., B.C., Z.H.Y., A.T.R. and M.A.K. participated in the discussions and interpretation of the data. All authors received and edited the manuscript.

## Competing interests

The authors declare no competing interests.
