## [Peer Review File · Nature Communications]

Signaling Mechanisms in Renal Compensatory Hypertrophy Revealed by Multi-OmicsREVIEWER COMMENTS

Reviewer #1 (Remarks to the Author):

In this manuscript, Kikuchi and colleagues utilize multiomics approaches to investigate the mechanism of compensatory kidney hypertrophy after uninephrectomy. Combining tubule microdissection with RNA-seq, ATAC-seq, proteomics and phosphoproteomics, they implicate Ppara in mediating the PT hypertrophic response. They validate this finding by activating Ppara with fenofibrate, which drove hypertrophy in the absence of uninephrectomy, and by abrogating PT hypertrophy after uninephrectomy in Ppara null mice. Finally, they show that collecting duct appears to react differently from PT, eg through hyperplasia via cell division, in contrast to PT.

These are well performed studies and the inclusion of proteomics is especially welcome. The data implicating Ppara as mediating the PT response are convincing and illustrate the power of this multiomic analysis.

Several issues should be addressed and/or clarified to improve the MS:

1. In contrast to the data supporting hypertrophy in PT, the data supporting hyperplasia in the CD are not as convincing – consisting of transcriptomic evidence and Li67 staining. Further validation should be performed to show that CD cells are actually completing mitosis – for example BrdU or EdU staining showing adjacent doublets that are positive at the 72 hour timepoint.
2. Ideally the authors would perform Ppara Cut&Run on control and hypertrophy cortex samples. It would be very informative to then intersect the CUT&RUN peaks with the ATAC peaks identified in the analysis.
3. It would be interesting to know whether the PT state induced by acute uninephrectomy has any similarity to any post-IRI states. The authors could test this by performing deconvolution analysis on published mouse IRI snRNA-seq datasets and looking for any change in cell type proportions represented by compensatory hypertrophy.
4. Performing a PPARa TF footprinting analysis of the ATAC-seq data would complement the motif enrichment analysis.
5. Can the authors link their ATAC peaks (especially those that change) to target genes using a published mouse snATAC-seq cis-coaccessibility analysis network?
6. Do the E2F_TARGETS and G2M_CHECKPOINT DEGs in 72hr PT and 24hr CCD overlap? Did PTs reach a conserved hyperplasia pathway once they went through the hypertrophy phase?
7. It is curious that gene expression vs Unx/sham promoter accessibility was highly correlated for PPARA but not HNF4A, so maybe some things to try:
8. Linking peaks to target genes: link distal HNF4A-binding peaks to target genes; does HNF4A target gene expression vs Unx/sham distal enhancer accessibility have significant correlation?
9. HNF4A might be enriched in all PT peaks at baseline; is there differential enrichment of PPARA/HNF4A motifs in differentially accessible peaks relative to all peaks?

Reviewer #2 (Remarks to the Author):

General comments

In their manuscript, Kikuchi et al present their multi-omics approach to elucidate the means and pathway by which compensatory hypertrophy after UNX takes place in different nephron segments. This is a very well-conducted study revealing differential (early) responses in the proximal tubula and collecting duct. It is a nice demonstration of a non-biased multi-omics

approach and provides valuable insights into the hypertrophic response in the kidney. Nevertheless, before this manuscript is suitable for publication, several issues need to be addressed.

Major comments

1: The title suggests that the authors have identified signalling events driving the adaptations after UNX. In my interpretation, this would refer to the stress response and the subsequent regulation of growth by intra- and/or extracellular signalling pathways. In line with their conclusion, replacing the term “signaling events” for “mechanisms” would better reflect the content of the paper.

2: Another word in the title that triggers me is “early”. In their introduction, the authors state that there is an immediate increase in GFR after UNX, “within minutes or hours” (line 62). Nevertheless, they opt to analyze kidneys at 24 hrs post UNX at the earliest, thereby, to my opinion, missing the real early signaling events that trigger the hypertrophic response. The authors should motivate their choice for the 24 h (and later) timepoints, and at least speculate on the events that initiate the response at timepoints as early as “minutes or hours”.

3: Supplemental Table 1 contains a lot of information and is referred to at several instances. I like to encourage the authors to provide a schematic representation of this table as one of the first figures in the main body of their manuscript.

4: I encourage the authors to include the RNA-seq data of both timepoints (24 and 72 hrs) in the figure presented in the main body.

5: For their proteomics studies, the authors used the entire kidney, stating that about 65% of the protein reflects proximal tubule (line 232). Although the authors have the capability to isolate specific nephron segments (and previously used these for proteomics analyses) they opt to analyze whole kidneys. Especially in the light of the study, where hypertrophy may affect the protein ratio, and different effects are observed for the proximal tubule and collecting duct, I find this a strange approach. Though I don't know the underlying considerations, I feel proteomic analyses of isolated segments would have been a better approach. It is not realistic to request this new analysis, but it would be fair if the authors could do the Western blot confirmations (as in figure 5) on isolated segments. Also they should discuss whether in a hypertrophic kidney also 65% of the protein is from the proximal tubule, or whether this ratio may shift.

Minor comments

1: Are ATAC- and RNA-seq data obtained from the same samples? Or separate samples? Please indicate.

2: The authors should provide illustrations and references for their statement in line 70; “Despite many reductionist...”.

3: The term ‘humoral’ (line 64), is that the best term here (regularly refers to an immunological response)

4: In figure 1C, the UNX contralateral kidney appears to have a rupture at the edge and have more open spaces in especially the cortex. Is this representative or are these artefacts, please elaborate.

5: in figure 2B: what is the reason some gene names are included, others not, what is the basis for this selection?

6: in line 155: “... the largest significant changes”.... Changes in what? Please specify here.

7: in line 171: add the term “identified” after “13,296”

8: line 210-212: “Therefore....tubules”: suggestion to rephrase sentence.

9: please check if all used abbreviations are properly introduced.

10: I don't see the need to refer to so many figures and suppl. Info in the discussion, this can be omitted.

Reviewer #3 (Remarks to the Author):

The paper by Kickuchi et al. analyzes the effect of uninephrectomy on kidney hypertrophy using an omic approach. The authors present an intriguing analysis of transcript and chromatin analysis, followed by proteomic analysis. They perform a multi-omic time course analysis that is of interest for the field. However, the following items need to be addressed before publication.

1. Hypertrophy is associated with metabolic alterations, in particular in lipid and amino acid metabolism and mitochondrial function.
 - mTOR, AMPK and the transcription factors HNF and PPAR should affect these processes, as well as the reabsorption processes of amino acids and sugars in the tubule. The metabolome is markedly absent from this dataset. Urinary/tissue measurements of key metabolites will need to be included to pinpoint the mechanisms of hypertrophy (and support hypothesis 1 as a key mechanism of hypertrophy).
 - Signals for amino acid metabolism and transport are markedly absent from the paper. One reason could be that the data presentation is selective.
 - While some of the mitochondrial functions show up (i.e. in the tubule RNAseq), a true integrative analysis across omics scales is missing. The authors should perform visualization and integration of inputs from each of the omics layers acquired.
2. Figure 2-4: The authors have several datasets and time points at hand. However, the data from the later time point (72h) is not visualized, and the last data point seems to be not analyzed at all. Better visualizations of the omics trajectories would increase the usefulness of this resource. Trajectories of early and late TF binding motifs if available should be visualized as well.
3. Figure 2: The ATAC-Seq data is interesting. However, there are several issues here that should be addressed with additional analyses.
 - The data lacks analysis of depth of chromatin sequencing and quality controls.
 - Is the single-tubule approach superior to comparable single-cell ATAC-seq approaches?
 - It is interesting to visualize the motifs in Figure 2C. However, many motifs usually appear in ATAC-seq and additional and FDR correction analyses need to be performed in order to investigate the priorities of transcription factor binding.
4. Fig. 4D needs nuclear costaining to confidently localize Ki67 to the nucleus. This Ki67 staining is not very convincing and does not appear to be in the nucleus. The conclusions regarding Ki67 positivity in the discussion cannot be supported at this time point.
5. The relevance of PPARalpha knockout is not clear. These mice have a long-described phenotype, with altered metabolism of many organs, including liver and kidney. The protective effect of fibrates has been shown in many experiments in kidney, including acute kidney injury (see for instance <https://pubmed.ncbi.nlm.nih.gov/14612380/>, <https://pubmed.ncbi.nlm.nih.gov/16316343/> <https://pubmed.ncbi.nlm.nih.gov/19710628/> and more from this and other groups. This probably needs to be added to the discussion.
6. Cell volume is lower with fenofibrate, but so is total kidney volume and body weight. To show that this is a kidney-specific effects, also other organs such as liver should be analyzed.
7. Translational importance of the paper is not clear to this reviewer.
8. Discussion: Some statements are oversimplified, i.e. the fate of membranes. As a means to increase the abundance of membranes, the authors state “a) transport into the cell; or (b) de novo synthesis in the cell”. At least three other mechanisms are: reduced shedding, repurposing of available membranes (i.e. endocytosis), reduced beta oxidation and lipolysis, and many more. In addition, membrane lipid composition is likely altered based on these inputs.

Point-by-point reply to editors and reviewers

We thank the editors and reviewers for their thoughtful comments, which have helped to improve the manuscript. We carried out additional studies to bolster our conclusions in accord with reviewer recommendations. We believe that we have successfully addressed the previously raised concerns. Our replies to the reviewers' comments are as follows

Reviewer(s)' Comments to Author:

Reviewer: 1

Comments to the author

In this manuscript, Kikuchi and colleagues utilize multiomics approaches to investigate the mechanism of compensatory kidney hypertrophy after uninephrectomy. Combining tubule microdissection with RNA-seq, ATAC-seq, proteomics and phosphoproteomics, they implicate Ppara in mediating the PT hypertrophic response. They validate this finding by activating Ppara with fenofibrate, which drove hypertrophy in the absence of uninephrectomy, and by abrogating PT hypertrophy after uninephrectomy in Ppara null mice. Finally, they show that collecting duct appears to react differently from PT, eg through hyperplasia via cell division, in contrast to PT.

These are well performed studies and the inclusion of proteomics is especially welcome. The data implicating Ppara as mediating the PT response are convincing and illustrate the power of this multiomic analysis.

Several issues should be addressed and/or clarified to improve the MS:

Response: Thank you. That may be a better summary than our own. We are very grateful for the constructive and thoughtful comments.

1. In contrast to the data supporting hypertrophy in PT, the data supporting hyperplasia in the CD are not as convincing – consisting of transcriptomic evidence and Ki67 staining. Further validation should be performed to show that CD cells are actually completing mitosis – for example BrdU or EdU staining showing adjacent doublets that are positive at the 72 hour timepoint.

Response: Thank you. To address these concerns, we switched the color for Ki 67 from red to green to increase the contrast, in parallel with the change of the color for AQP2 from green to red. Subsequently, to clearly illustrate the localization of Ki67 in the nucleus, we prepared a figure with co-staining of Ki67 and DAPI with higher magnification (**Figure 5D**). Although we performed additional staining for PCNA to validate this finding, as is shown in **Figure R1** below, staining of PCNA was too faint to be included in the figure for publications.

Figure. R1 Immunofluorescence labeling of mouse renal cortex 72 hours after UNx. Labelling for aquaporin-2 (AQP2; red) identifies collecting ducts. PCNA (green) identifies dividing cells.

2. Ideally the authors would perform Ppara Cut&Run on control and hypertrophy cortex samples. It would be very informative to then intersect the CUT&RUN peaks with the ATAC peaks identified in the analysis.

Response: We agree that PPAR α binding site data would be a valuable addition. We have begun performing these experiments based on the reviewer's suggestion. However, the optimization required has placed these data outside this manuscript's review time frame. Thus, we will include these data in a separate manuscript in the future. In the meantime, the use of PPAR α binding motifs to infer PPAR α target sites has produced valuable insights in the current manuscript.

3. It would be interesting to know whether the PT state induced by acute uninephrectomy has any similarity to any post-IRI states. The authors could test this by performing deconvolution analysis on published mouse IRI snRNA-seq datasets and looking for any change in cell type proportions represented by compensatory hypertrophy.

Response: To perform deconvolution analysis, we compared UNx-responsive genes from our data set with genes annotated as "Healthy S1" or "Injured/Severe injured PT" in Kirita's previous study (PMID: 31506348). From this analysis, we can see some pattern showing that "Healthy-PT" related genes tend to be decreased in UNx kidney (**Figure R2-A, Supplementary Fig. 8A**), and "Injured-PT" related genes are both increased and decreased in UNx kidney (**Figure R2-B, Supplementary Fig. 8B**). However, the sensitivity of this analysis was not ideal: most differentially expressed genes ($p < 0.1$) in our study have

small PCT.1 values (fraction of cells expressing the genes in the selected cells) in the “Healthy PT” or “Injured PT” gene sets from the IRI snRNA-seq study (**Figure R2-C and D, Supplementary Fig. 8C and D**). This indicates difficulty in associating our renal hypertrophy model with the acute renal injury model. We added and discussed the results of this analysis to **Supplementary discussion 2** and **Supplementary figure 8**.

Figure. R2 (Supplementary Figure 8)

Deconvolution analysis for PT-S1 RNA seq dataset using previous mouse IRI snRNA-seq dataset (Proc Natl Acad Sci U S A. 2019 Sep 24;116(39):19619-19625.PMID: 31506348)

4. Performing a PPAR α TF footprinting analysis of the ATAC-seq data would complement the motif enrichment analysis.

Response: We have included a new panel in **Figure 3** with the requested analysis. In short, we used all detected peaks in our ATAC data to perform transcription factor footprinting for PPAR α to visualize the relationship between PPAR α motifs and chromatin accessibility. We observed a well-defined footprint immediately surrounding PPAR α binding motifs (**Figure 3D-Top**). UNx samples showed increased chromatin accessibility surrounding PPAR α motifs, while UNx samples did not show any difference of chromatin accessibility surrounding CTCF motifs (**Figure 3D-Bottom**)

5. Can the authors link their ATAC peaks (especially those that change) to target genes using a published mouse snATAC-seq cis-coaccessibility analysis network?

Response: Thank you for the suggestion.

Previously, peaks in mouse having significant correlation between their target genes in the kidney (n = 1260) were reported (Science 2018, Cao J, Darren A.C, Cole T et al (PMID: 30166440)). We intersected these peaks with the DARs detected in our bulk ATAC seq to determine whether our DAR regions could be associated with expression of gene targets.

Due in part to the small peak set size from the Cao et al. paper, we found only 30 DARs that intersected with peaks reported to have significant correlation to their target gene expression. However, all 30 DARs have weak correlation coefficient values in their data (**Figure R3**).

We visually inspected these 30 intersecting peak regions to determine if the PPAR α binding motif was present using the UCSC genome browser (**Figure R3**). Among these, only a few regions contained PPAR α motifs in their peak locations. For example, the region of chr14:54754897-54755118, in which PPAR α motif was present, was reported to have correlation to the gene expression of *Slc7a8*. Although speculative, altered accessibility of this region might allow PPAR α to access this binding site, which could mediate the change of gene expression of *Slc7a8* (PMID 18489776). Also, it might be possible that sparsity of PPAR α motifs in these regions is one of the reasons why the correlation coefficients found in these 30 DARs are very small (average = 0.07). In either case, it is our view that this analysis does not warrant inclusion in the main manuscript due to the very small size of this peak intersection.

6. Do the E2F_TARGETS and G2M_CHECKPOINT DEGs in 72hr PT and 24hr CCD overlap? Did PTs reach a conserved hyperplasia pathway once they went through the hypertrophy phase?

Response: Thank you. Upregulated genes associated with “G2M checkpoint” and “E2F targets” in GSEA substantially overlapped with those seen in CCD at the 24 hour timepoint. We summarized this analysis in **Supplementary Data 14** and added the figure showing this overlap in Venn diagrams in **Figure 6D**.

7. It is curious that gene expression vs Unx/sham promoter accessibility was highly correlated for PPARA but not HNF4A, so maybe some things to try:

Thank you.

8. Linking peaks to target genes: link distal HNF4A-binding peaks to target genes; does HNF4A target gene expression vs Unx/sham distal enhancer accessibility have significant correlation?

Response: Thank you for the insightful comment. No significant relationship was found between ATAC-seq peaks in distal enhancer regions (Intergenic and Intronic regions) for either PPAR α or HNF α target genes. This finding was added as **Supplementary Fig. 3D**.

9. HNF4A might be enriched in all PT peaks at baseline; is there differential enrichment of PPARA/HNF4A motifs in differentially accessible peaks relative to all peaks?

Response: Thank you. Differentially accessible regions exhibited sharply elevated enrichment of the HNF4 α /PPAR α binding motif compared to the total peak set (47% vs. 28%). In contrast, no enrichment was found for the HNF1B motif (negative control). This finding was added in **Figure 3C**.

Reviewer #2 (Remarks to the Author):

General comments

In their manuscript, Kikuchi et al present their multi-omics approach to elucidate the means and pathway by which compensatory hypertrophy after UNX takes place in different nephron segments. This is a very well-conducted study revealing differential (early) responses in the proximal tubula and collecting duct. It is a nice demonstration of a non-biased multi-omics approach and provides valuable insights into the hypertrophic response in the kidney. Nevertheless, before this manuscript is suitable for publication, several issues need to be addressed.

Response: We are very grateful for your constructive and thoughtful review of our manuscript. The itemized responses to your comments are below.

Major comments

1: The title suggests that the authors have identified signalling events driving the adaptations after UNX. In my interpretation, this would refer to the stress response and the subsequent regulation of growth by intra- and/or extracellular signalling pathways. In line with their conclusion, replacing the term “signaling events” for “mechanisms” would better reflect the content of the paper.

Response: Thank you. As was suggested, we changed the title to “Signaling Mechanisms in Renal Compensatory Hypertrophy Revealed by Multi-Omics”.

2: Another word in the title that triggers me is “early”. In their introduction, the authors state that there is an immediate increase in GFR after UNX, “within minutes or hours” (line 62). Nevertheless, they opt to analyze kidneys at 24 hrs post UNX at the earliest, thereby, to my opinion, missing the real early signaling events that trigger the hypertrophic response. The authors should motivate their choice for the 24 h (and later) timepoints, and at least speculate on the events that initiate the response at timepoints as early as “minutes or hours”.

Response: Thank you. As was suggested, we removed the term “Early” from the title and throughout the manuscript. As you pointed out, single-nephron GFR increases within minutes or hours after UNx. Based on our lipid analysis that found substantially elevated levels of lipid ligands of PPAR α at 24 hours after UNx, we agree that the activation of PPAR α is likely to also be detectable at time points earlier than 24 hours. This point was added to the discussion section (**Page 18, Paragraph 2, lines 4-6**).

3: Supplemental Table 1 contains a lot of information and is referred to at several instances. I like to encourage the authors to provide a schematic representation of this table as one of the first figure in the main body of their manuscript.

Response: Thank you. We created a new schematic representation of the hypotheses. Please see **Figure 2**.

4: I encourage the authors to include the RNA-seq data of both timepoints (24 and 72 hrs) in the figure presented in the main body.

Response: Thank you. To address this comment, we separated the 24-hour and 72-hour RNA-seq data for proximal tubule into two main figures (**Figure 4** for 24 hour RNA-seq, **Figure 6** for 72 hour RNA-seq).

5: For their proteomics studies, the authors used the entire kidney, stating that about 65% of the protein reflects proximal tubule (line 232). Although the authors have the capability to isolate specific nephron segments (and previously used these for proteomics analyses) they opt to analyze whole kidneys. Especially in the light of the study, where hypertrophy may affect the protein ratio, and different effects are observed for the proximal tubule and collecting duct, I find this a strange approach. Though I don't know the underlying considerations, I feel proteomic analyses of isolated segments would have been a better approach. It is not realistic to request this new analysis, but it would be fair if the authors could do the Western blot confirmations (as in figure 5) on isolated segments. Also they should discuss whether in a hypertrophic kidney also 65% of the protein is from the proximal tubule, or whether this ratio may shift.

Response: Thank you for thoughtful comments. We used whole kidney in the original proteomics experiment because we wanted to perform phospho-proteomics in parallel with total proteomics experiments. Given that phosphorylated proteins account for only 1 % of total protein, it was technically impossible to perform phospho-proteomics from micro-dissected tubules due to the small sample volume. Also, microdissection takes normally at least 1 hour, which might make it difficult to detect changes caused by hypertrophy given that the sensitivity of proteomics is much weaker than RNA-seq.

To address the reviewer's comments, we performed a new proteomics experiment using isolated kidney cortices, of which 84% of total proteins is derived from proximal tubules (vs. 65 percent in whole kidney (PMID: 31253652), now described in the revised paper). As is shown in **Supplementary Fig. 6F**, we observed substantial de-enrichment of the collecting-ducts (AQP2) fraction in kidney cortex compared to whole kidney. These new kidney cortex proteomics data are shown in **Figure 7F**, and the whole data set is available in **Supplementary Data 18**.

One wonders, "Could the changes in individual proteins be due to a change in the percent of proximal tubule?" Even if the percent of proximal tubule proteins in cortex increased from 84% percent to 100%, that would only be a 16% increase in proximal tubule proteins, which is dwarfed by the changes actually seen. We believe the addition of the kidney cortex proteomics analysis in the revised paper improves our level of confidence in the conclusions and has strengthened the manuscript.

Minor comments

1: Are ATAC- and RNA-seq data obtained from the same samples? Or separate samples? Please indicate.

Different samples.

We added "newly prepared" to the sentence as in below. (**Page 8, Paragraph 4**)

"we used RNA-seq in newly prepared S1 proximal tubules micro-dissected from contralateral mouse kidneys"

2: The authors should provide illustrations and references for their statement in line 70; "Despite many reductionist...".

Thank you. We added information regarding several previous important investigations showing the mechanisms of compensatory renal hypertrophy.

3: The term 'humoral' (line 64), is that the best term here (regularly refers to an immunological response)

Thank you. Changed to “Circulating factor”.

4: In figure 1C, the UNX contralateral kidney appears to have a rupture at the edge and have more open spaces in especially the cortex. Is this representative or are these artefacts, please elaborate.

Thank you. We revised this figure from images of HE-staining to that of picture from dissection scope (**Figure 1C**).

5: in figure 2B: what is the reason some gene names are included, others not, what is the basis for this selection?

Thank you. We elaborated the volcano plot for ATAC-seq (Promoter-TSS region, **Figure 3E**).

Chromatin accessible regions with top 10 \log_2 (UNx/Sham) and bottom 10 \log_2 (UNx/Sham) were annotated by nearest gene name (Homer Annotate).

6: in line 155: “ ... the largest significant changes” Changes in what? Please specify here.

Thank you. The average of \log_2 (UNx peak concentration/Sham peak concentration) for all identified regions. Fixed.

7: in line 171: add the term “identified” after “13,296”

Fixed.

8: line 210-212: “Therefore....tubules”: suggestion to rephrase sentence.

Thank you. Fixed.

9: please check if all used abbreviations are properly introduced.

Thank you. Fixed.

10: I don't see the need to refer to so many figures and suppl. Info in the discussion, this can be omitted.

Thank you. Fixed.

Reviewer #3 (Remarks to the Author):

The paper by Kickuchi et al. analyzes the effect of uninephrectomy on kidney hypertrophy using an omic approach. The authors present an intriguing analysis of transcript and chromatin analysis, followed by proteomic analysis. They perform a multi-omic time course analysis that is of interest for the field. However, the following items need to be addressed before publication.

Response: We are very grateful for your constructive and thoughtful review of our manuscript.

1. Hypertrophy is associated with metabolic alterations, in particular in lipid and amino acid metabolism and mitochondrial function.

- mTOR, AMPK and the transcription factors HNF and PPAR should affect these processes, as well as the reabsorption processes of amino acids and sugars in the tubule.

The metabolome is markedly absent from this dataset. Urinary/tissue measurements of key metabolites will need to be included to pinpoint the mechanisms of hypertrophy (and support hypothesis 1 as a key mechanism of hypertrophy).

Response: Thank you for thoughtful comments. As was pointed out, PPAR α endogenous ligands are saturated and unsaturated fatty acids or their derivatives as well as eicosanoids and arachidonic acid metabolites. (Mol. Endocrinol., 1997, 11, 779-791, Nature, 1996, 384, 39-43. Biochemistry (Mosc) 2005, 44(4):1193–1209.) Therefore, we agreed that the analysis of abundance of the fatty acid metabolites in the kidney would be insightful for our study.

To address this question, we collaborated with Dr. Alan T. Remaley (Lipoprotein Metabolism Laboratory, NHLBI), an expert in lipidomics.

We performed GC analysis of mouse whole kidney tissue and observed significant increases in absolute abundances of SAT, MUFA, PUFA in the hypertrophied kidney obtained at 24 hours post-UNx. This result strongly supports our hypothesis that increased single nephron GFR results in elevated levels of the fatty acid ligands of PPAR α in mouse kidney after UNx. The results of this lipid analysis are presented in the new **Figure 9C and D**. In addition, the quantitative analysis of lipid content in mouse kidney (**Figure 9D and E, Supplementary Data 27**) revealed that the concentration of triglycerides and phospholipids in the kidney after UNx were increased by ~2-fold and 80.8%, respectively, compared with the Sham group. This observation is consistent with the increased levels of overall fatty acid concentrations in the UNx group.

With regard to the amino-acid metabolome in the kidney tissue, although we could not perform metabolomics analysis, a previous study showed that the concentrations of free methionine, alanine, tyrosine, valine and leucine in renal cortical tissue were increased two days after unilateral nephrectomy in young male Sprague-Dawley rats, as detected by ion-exchange chromatography. (THE YALE JOURNAL OF BIOLOGY AND MEDICINE 51 (1978), 395-401). Thus, we added this point to the discussion section. (**Page 20, Paragraph 2, Lines 3-6**)

- Signals for amino acid metabolism and transport are markedly absent from the paper. One reason could be that the data presentation is selective.

To address this, we performed data integration to *systematically* identify biological processes most likely to regulate renal hypertrophy in proximal tubules. In short, results from the nine -omics data sets presented in this study (**Figure 9A**) were integrated using the Database for Annotation, Visualization and Integrated Discovery (DAVID) (<https://david.ncifcrf.gov/home.jsp>) (**Figure 9A, Supplementary Data 25**).

In short, all \log_2 values from each nine omics datasets (ATAC-seq for TSS-promoter regions, ATAC-seq for Intergenic regions, RNA seq for proximal tubule S1 segment at 24 hour, RNA seq for proximal tubule S1 segment at 72 hour, proteomics for whole kidney at 24 hour, proteomics for kidney cortex at 24 hour, proteomics for whole kidney at 72 hour, phosphoproteomics for whole kidney at 24 hour and phosphoproteomics for whole kidney at 72 hour) were normalized into z scores to determine differential enrichment or expression status for each ATAC-peak, gene or protein ($z > 1.96$, or $z < -1.96$) between UNx and sham treatments. Differentially enriched/expressed factors were assigned the logical score =1, while unchanged factors were assigned a logical score = 0. Finally, all logical scores were summed up, and genes (proteins) with a total logical score more than 3 were included in the DAVID analysis to identify enriched Gene Ontology Biological Processes.

The results of the gene ontology analysis predominantly highlighted changes in fatty acid metabolism including “lipid transport”, which is consistent with PPAR α activation as found in this study (**Figure 9B**). In contrast, the \log_2 values for the amino acid transporters reported to be expressed in the kidney (Am J Physiol Cell Physiol. 2021 Sep 1;321(3):C507-C518.) were not substantially altered at either time point. Among these, only the glutamine transporter *Slc38a3* showed significant increase at 24 hours (**Supplementary Data 26**).

To further address the reviewer’s concerns, we performed multiple new analyses on the sets of genes/proteins annotated as “MTORC1 signaling” in the GSEA database. These include a Chi-square analysis (**Supplementary Fig. 3B-center, Figure 7B-center, Figure 8B-right**), a trajectory analysis showing differential changes between the 24 and 72 hour timepoints (**Supplementary Fig 5B-center, Figure 8C-right**), and correlations between the 24-hour proteomics data and the 24-hour RNA seq data (**Supplementary Fig. 6B-center**).

- While some of the mitochondrial functions show up (i.e. in the tubule RNAseq), a true integrative analysis across omics scales is missing. The authors should perform visualization and integration of inputs from each of the omics layers acquired.

Response: Thank you. To address the comments, we performed an integrative analysis across the nine omics data sets and visualized the results (**Figure 9A, and B**) as described above.

In addition, figures including a Chi-square analysis for genes/proteins annotated as “OXIDATIVE PHOSPHORYLATION” in GSEA (**Supplementary Fig. 3B-right, Figure 7B-right**), a trajectory analysis for curated genes/proteins annotated as “OXIDATIVE PHOSPHORYLATION” showing differential changes between the 24 and 72 hour timepoints (**Supplementary Fig 5B-right**), and correlations between 24 hour proteomics data and 24 hour RNA seq data for genes/proteins annotated as “OXIDATIVE PHOSPHORYLATION” (**Supplementary Fig. 6B-right**) are provided in the revised manuscript.

2. Figure 2-4: The authors have several datasets and time points at hand. However, the data from the later time point (72h) is not visualized, and the last data point seems to be not analyzed at all. Better

visualizations of the omics trajectories would increase the usefulness of this resource. Trajectories of early and late TF binding motifs if available should be visualized as well.

Response: Thank you. First, as was suggested, we moved the figures for 72 hour RNA seq and 72 hour proteomics to the main figures. In order to highlight the differential changes between the 24 and 72 hour timepoints, we added bubble plots for the RNA-seq and proteomics data sets (**Figure 8C, Supplementary Fig 5B**).

3. Figure 2: The ATAC-Seq data is interesting. However, there are several issues here that should be addressed with additional analyses.

- The data lacks analysis of depth of chromatin sequencing and quality controls.

Response: Thank you. We added the figures for quality control of ATAC-seq including a heatmap for the distribution of reads around TSS and a fragment size distribution plot showing enriched nucleosome-free fragments. The heatmap illustrated the enriched nucleosome-free fragments, and the fragment size distribution plot shows enrichment around 100 and 200 bp, which indicates nucleosome-free and mono-nucleosome-bound fragments. In addition, a table was added to summarize additional quality-control metrics such as non-redundant fraction (NRF), the fraction of reads in called peak regions (FRiP score), and read depths after removal of duplicates. (**Supplementary Fig 2C**)

- Is the single-tubule approach superior to 12 comparable single-cell ATAC-seq approaches?

Response: Thank you. Single-cell techniques are vital when investigating cell-to-cell variations in phenotypes. However, we do not focus on phenotypes at the level of individual cells in this study. Further, single-cell -omics methods often provide lower sensitivity than comparable bulk-cell methods. Thus, for our purposes of comparing PT-status between UNx vs. Sham, bulk-cell methods are more suitable (provided that proximal tubule segments can be dissected out robustly). In addition, specific nephron segments such as PT-S1 can be obtained directly by eye during micro-dissection, whereas the use of genomic cellular markers would be needed for such tissue identification in a single-cell -omics experiment. The following snippet from a recent talk highlights advantage of single-tubule over single cell (**Figure R4**).

Limitations of Single-Cell RNA-Seq

Single-Cell RNA-Seq

- Cell stress (loss of cell-cell junctions)
- Limited viability of some cell types
- Drop out cells
- Lack of data-independent classification
- Limited depth
- Loss of isoform-specific information

Single-Tubule RNA-Seq

- Epithelial structure remains intact
- Fast
- No drop outs
- Unambiguous identification of segments
- Comprehensive transcriptomes
- Isoform-specific quantification

Figure. R4 Comparison of single-cell RNA-seq vs Single-tubule RNA-seq

Dropouts is the cells that express a given protein, but do not contain measurable amounts of the corresponding mRNA

- It is interesting to visualize the motifs in Figure 2C. However, many motifs usually appear in ATAC-seq and additional and FDR correction analyses need to be performed in order to investigate the priorities of transcription factor binding.

Response : Thank you. We performed FDR corrections to calculate q-values. All data are available in **Supplementary Data 5**. Also, figures are corrected, accordingly (**Figure 3B, Supplementary Figure 2D**).

4. Fig. 4D needs nuclear costaining to confidently localize Ki67 to the nucleus. This Ki67 staining is not very convincing and does not appear to be in the nucleus. The conclusions regarding Ki67 positivity in the discussion cannot be supported at this time point.

Response: Thank you. We agree that the quality of Figure of Ki67 was not good enough to be used as a representative figure. To address these concerns, we have substantially modified this figure as described in the response to Reviewer #1 above. (**Figure 5D**)

5. The relevance of PPAR alpha knockout is not clear. These mice have a long-described phenotype, with altered metabolism of many organs, including liver and kidney. The protective effect of fibrates has been shown in many experiments in kidney, including acute kidney injury (see for instance <https://pubmed.ncbi.nlm.nih.gov/14612380/>, <https://pubmed.ncbi.nlm.nih.gov/16316343/> <https://pubmed.ncbi.nlm.nih.gov/19710628/> and more from this and other groups. This probably needs to be added to the discussion.

Response: Thank you for the suggestion. We discussed this considering translational importance of this paper in the **Supplementary discussion 2**.

6. Cell volume is lower with fenofibrate, but so is total kidney volume and body weight. To show that this is a kidney-specific effects, also other organs such as liver should be analyzed.

Response: Thank you for insightful comments. We added a cell size assay for the liver and found that the effect of fenofibrate is not kidney-specific, but systemic (**Supplementary Fig, 7D, Supplementary Data 30**).

7. Translational importance of the paper is not clear to this reviewer.

Response: Thank you for the suggestion. To know whether the proximal tubule status induced by UNx is relevant to post-acute renal injury model, we performed a deconvolution analysis for our PT-S1 RNA seq dataset at 24 hours using a published mouse IRI snRNA-seq datasets (Proc Natl Acad Sci U S A. 2019 Sep 24;116(39):19619-19625.PMID: 31506348). We looked for any change in cell type proportions represented by compensatory hypertrophy (**Supplementary Fig. 8**). From this analysis, we can see some pattern showing that "Healthy-PT" related genes tend to be decreased in UNx kidney (**Supplementary Fig. 8A**), and "Injured-PT" related genes are both increased and decreased in UNx kidney. (**Supplementary Fig. 8B**). However, to be noted, most differentially expressed genes ($p < 0.1$) in our study annotated as either "Healthy PT" or "Injured PT" have small PCT.1 values (fraction of cells expressing the genes in the selected cells) as defined in the mouse IRI snRNA-seq study (**Supplementary Fig. 8C and D**), indicating difficulty in associating our renal hypertrophy model with this acute renal injury model. Future studies would benefit from deconvolution analysis of our study on single cell RNA-

seq using other kidney disease models, such as cisplatin-induced acute renal failure model, mentioned in question 5 (**Supplementary Discussion 2**).

8. Discussion: Some statements are oversimplified, i.e. the fate of membranes. As a means to increase the abundance of membranes, the authors state “a) transport into the cell; or (b) de novo synthesis in the cell”. At least three other mechanisms are: reduced shedding, repurposing of available membranes (i.e. endocytosis), reduced beta oxidation and lipolysis, and many more. In addition, membrane lipid composition is likely altered based on these inputs.

Thank you for thoughtful comments. We fixed the sentence.

REVIEWERS' COMMENTS

Reviewer #1 (Remarks to the Author):

This is a responsive review that includes new metabolomic and microdissected tubule analyses.

Reviewer #2 (Remarks to the Author):

The authors have adequately addressed my comments, there are just two minor items that remain:

In response to my "Major comment 2" the authors state "we agree that the activation of PPAR α is likely to also be detectable at time points earlier than 24 hours. This point was added to the discussion section (Page 18, Paragraph 2, lines 4-6)", while at this point they state "it is POSSIBLE that the initial activation of PPAR α occurs earlier than 24 hours post-UNx.". To be honest... I prefer "likely" over "possible".

Additionally, I suggest the authors to check Figure 7B and supplemental Figure 6B and references/legends to these figures for the spelling of the word "Oxidative" (now: oxiTative, with a T)

Reviewer #3 (Remarks to the Author):

The authors have done a great job in adressing my comments.

Point by point responses to reviewer comments:

Reviewer #1:

Reviewer: This is a responsive review that includes new metabolomic and microdissected tubule analyses.

Response: Thanks for helping us improve this manuscript with your many constructive suggestions.

Reviewer #2:

Reviewer: The authors have adequately addressed my comments, there are just **two minor items** that remain.

Response: Thanks. See below.

Reviewer: In response to my “Major comment 2” the authors state “we agree that the activation of PPAR α is likely to also be detectable at time points earlier than 24 hours. This point was added to the discussion section (Page 18, Paragraph 2, lines 4-6)”, while at this point they state “it is POSSIBLE that the initial activation of PPAR α occurs earlier than 24 hours post-UNx.”. To be honest... I prefer “**likely**” over “**possible**”.

Response: We have made this change.

Reviewer: Additionally, I suggest the authors to check **Figure 7B** and supplemental Figure 6B and references/legends to these figures for the spelling of the word “Oxidative” (now: oxiTative, with a T)

Response: fixed. Thanks.

Reviewer #3:

Reviewer: The authors have done **a great job** in addressing my comments.

Response: Detailed reviews like the ones we got from the three referees are important elements of the publication process. Thanks for the time spent to help us improve our paper.